# Resolving Partial Observability in Decision Processes via the Lambda Discrepancy

## Abstract

We consider the reinforcement learning problem under partial observability, where observations in the decision process lack the Markov property. To cope with partial observability, first we must detect it. We introduce the $\lambda$-discrepancy: a measure of the degree of non-Markovianity of system dynamics. The $\lambda$-discrepancy is the difference between TD($\lambda$) value functions for two different values of $\lambda$; for example, between 1-step temporal difference learning (TD(0)), which makes an implicit Markov assumption, and Monte Carlo value estimation (TD(1)), which does not. We prove that this observable and scalable value-based measure is a reliable signal of partial observability. We then use it as an optimization target for resolving partial observability by searching for memory functions—functions over the agent's history—to augment the agent's observations and reduce $\lambda$-discrepancy. We empirically demonstrate that our approach produces memory-augmented observations that resolve partial observability and improve decision making.

## 1 Introduction

Reinforcement learning (Sutton and Barto, 1998) (or *RL*) tasks are typically assumed to be well-modeled as Markov decision processes (or *MDPs*), where an agent interacting with a task observes necessary information required to fully describe the state of the task. This assumption—the *Markov assumption*—is virtually ubiquitous in RL research, and much of the theoretical framework upon which RL algorithms are built depends upon it. However, decision processes in the real world are rarely Markov, and expert effort is typically required to artificially augment real-world observation spaces to render them Markov. For example, when RL is applied to Atari (Bellemare et al., 2013), the current observation is stacked with three previous frames to include information about the velocity and acceleration of sprites (Mnih et al., 2015). Applications from automated HVAC control (Galataud, 2021) to stratospheric balloon navigation (Bellemare et al., 2020) use task-specific state features hand-engineered to approximate the Markov assumption and incorporate history information. While these applications are promising, the hand-designed features lack generality across tasks.

General decision-making agents must overcome non-Markovian observations without the benefit of external intervention. Here, the dominant problem model is the *partially observable* Markov decision process (or *POMDP*), which relaxes the Markov assumption to hold only for unobserved states. The environment dynamics, reward function, and the agent's observations all depend on these underlying states, but observations may not be Markovian themselves, and—in contrast to MDPs—optimal behavior is usually history-dependent. Memory can reestablish or approximate the Markov property by summarizing history, but few approaches offer practical guidance on what to remember and when. Often, the agent's memory is simply optimized concurrently with the same learning signal as the value function or policy.

The optimization of value and memory need not be tackled simultaneously: with an appropriate learning signal, memory can be optimized independently from value and policy. Separating these objectives adds algorithmic flexibility and allows memory learning to benefit off-policy evaluation and offline as well as online reinforcement learning. Here we focus solely on memory optimization. We propose a measure of partial observability in general decision processes that we call the $\lambda$-*discrepancy*: the difference between two different value functions estimated using the temporal difference learning procedure TD($\lambda$) (Sutton, 1988). In TD($\lambda$), the parameter $\lambda$ trades off between short-term and long-term Bellman backups, where $\lambda = 0$ corresponds to 1-step backups, and $\lambda = 1$

uses infinite backups. The latter case, which is equivalent to estimating the Monte-Carlo (MC) return, is the only unbiased choice of $\lambda$ in the partially observable setting. All other choices introduce bias in the value function fixed-point due to the implicit Markov assumption contained in the Bellman operator. We show that—for non-Markov decision processes only—this value discrepancy reliably exists for any two parameter choices $\lambda_1 \neq \lambda_2$ and almost all policies. Our work investigates using this measure to both detect and resolve partial observability.

We make the following contributions:

1. We analyze the sources of partial observability in general decision processes, and introduce the $\lambda$-discrepancy, a measure of non-Markovianity.
2. We prove that the $\lambda$-discrepancy can be used to reliably identify non-Markovian reward or transition dynamics.
3. We then consider a "best-case" optimization scheme that adjusts the parameters of a memory function to minimize the $\lambda$-discrepancy via gradient descent. This approach requires computing the $\lambda$-discrepancy in closed form, and is therefore only feasible in the planning setting. However, the results demonstrate that minimizing $\lambda$-discrepancy reduces partial observability and improves agent performance across a range of classic POMDPs.
4. Finally, we consider the more realistic setting where the agent only has point estimates of the $\lambda$-discrepancy. We equip the agent with oracle value functions and search for a $\lambda$-discrepancy-minimizing memory function using hill-climbing. The resulting memory functions achieve nearly the same final performance as the best-case gradient-based method.

Based on our theoretical and empirical analysis, we conclude that the $\lambda$-discrepancy is a reliable and useful measure for detecting and reducing partial observability in decision processes. It is also practical, since the $\lambda$-discrepancy can be computed directly from value functions, which many reinforcement learners already estimate anyway. Furthermore, since the $\lambda$-discrepancy is a function of only *observable* quantities in a decision process, it is feasible to apply it in problems where state information is truly hidden, and where the agent may not even *know* the full set of possible states.

## 2 BACKGROUND

In the typical RL model, an agent tries to maximize expected rewards in an MDP (Puterman, 1994). An MDP is defined by the tuple $(S, A, R, T, \gamma, p_0)$, where $S$ is the state space, $A$ is the action space, $R : S \times A \to \mathbb{R}$ is the reward function, $T : S \times A \times S \to [0, 1]$ is the transition function, $\gamma \in [0, 1]$ is the discount factor, and $p_0 : S \to [0, 1]$ is the start-state distribution. The agent's goal is to find a policy $\pi$ that selects actions to maximize *return*, $G_t$, the discounted sum of future rewards starting from time step $t$: $G_t^\pi = \sum_{i=0}^{\infty} \gamma^i r_{t+i}$, where $r_i$ is the observed reward at time step $i$. We denote the expectation of these returns as *value functions* $V_\pi(s_t) = \mathbb{E}_\pi[G_t \mid S_t = s_t]$ and $Q_\pi(s_t, a_t) = \mathbb{E}_\pi[G_t \mid S_t = s_t, A_t = a_t]$.

A decision process is Markovian if it satisfies two properties, one describing $T$ and one $R$. In an MDP, the state $s_t \in S$ and action $a_t \in A$ at time step $t$ together are a sufficient statistic for predicting state $s_{t+1}$ and reward $r_t$, instead of requiring the agent's whole history:

$$T(s_{t+1}|s_t, a_t) = T(s_{t+1}|s_t, a_t, \ldots, s_0, a_0); \ \ R(s_t, a_t) = R(s_t, a_t, \ldots, s_0, a_0). \tag{1}$$

The Markov assumption has several desirable implications. First, the transition and reward functions, and consequently the value functions $V_\pi(s)$ and $Q_\pi(s, a)$, have fixed-sized inputs and are therefore easy to parameterize, learn, and reuse. Second, it follows that the optimal policy $\pi^* : S \to A$ need only reactively and deterministically map from states to actions. Finally, if the Markov property holds then so does the Bellman equation:

$$V_\pi(s_t) = \mathbb{E}_\pi \left[ R_t + \gamma V_\pi(S_{t+1}) \right], \tag{2}$$

where $S_{t+1}$ and $R_t$ are the state and reward at the respective time steps. Equation 2 defines a recurrence relation over expected one-step returns. We can unroll this relation to obtain a version for $n$-step returns: $V_\pi(s_t) = \mathbb{E}_\pi[G_{t:t+n}]$, where $G_{t:t+n} \doteq R_t + \gamma R_{t+1} + \gamma^2 R_{t+2} + \cdots + \gamma^n V_\pi(S_{t+n})$. The same equation holds for weighted combinations of such returns, including the exponential average:

$$V_\pi^\lambda(s_t) = \mathbb{E}_\pi \left[ (1 - \lambda) \sum_{n=1}^{\infty} \lambda^{n-1} G_{t:t+n} \right]. \tag{3}$$

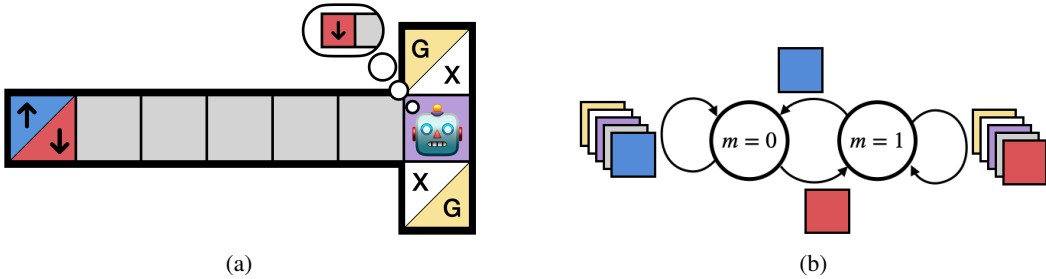

(a)                                         (b)

Figure 1: (a) Bakker's T-Maze, an example of a non-Markov decision processes. The agent must navigate through the environment but can only observe the color of the square it is currently in. The agent will more easily reach the goal (G) and receive positive reward if it can remember at the purple junction whether it previously saw a blue or a red observation. G and X are terminal states. (b) A learnt memory function that tests whether a blue or a red observation was observed more recently.

The argument of this expectation is known as the $\lambda$-return (Sutton, 1988), and the expectation itself defines the TD($\lambda$) value function. Due to the Bellman equation, all TD($\lambda$) value functions for a given MDP and policy share the same fixed point for any $\lambda \in [0, 1]$.

In the general case, where the Markov property does not hold, the resulting *decision process* is described by a tuple $(\Omega, A, R, T, \gamma)$. In this setting, the agent receives observations $\omega \in \Omega$ at each time step $t$ instead of states, and all the relevant functions—transition $T$ and reward $R$, and therefore policy $\pi$ and value functions $V$ and $Q$—are functions of the agent's entire history $h_t$. Frequently, such decision processes are modeled as *partially-observable MDPs* (POMDPs) (Kaelbling et al., 1998), where transitions and rewards are defined over an unobserved latent state that generates the agent's observations. Rather than define history-dependent $T$ and $R$, the POMDP model extends the basic MDP by adding observations $\Omega$ and an observation function $O(\omega|s)$ representing the probability of seeing observation $\omega$ in latent state $s$. Typically, observations $\omega$ do not contain enough information to fully resolve states $s$, and consequently, the Bellman equation no longer holds, and different $\lambda$ may have different TD($\lambda$) fixed points. In this work, we seek to characterize this phenomenon, as well as exploit it to detect and resolve partial observability by measuring differences in the fixed points of TD($\lambda$) value functions.

## 3  A Measure of Partial Observability

The two dominant reinforcement learning paradigms, MDPs and POMDPs, start by assuming either fully or partially observable state information. Rather than adopt one of these two decision making paradigms wholesale, we instead advocate for inspecting the decision process to detect and resolve partial observability whenever it affects value. By systematically using memory to better approximate the Markov assumption, we can enable more effective decision making in general decision processes.

Let us first examine why the Markov property does not hold for observations in general decision processes. Consider the T-Maze example of Figure 1a. The initial observation (BLUE/RED) indicates the location of the rewarding goal state G (UP/DOWN, respectively). However, since the agent can only observe the color of its current grid cell, the gray corridor and purple junction provide no information about the goal. The agent must use the information in the starting square to select actions effectively at the junction square.

Recall that under the Markov assumption (Equation 1), additional history does not add precision; the transition and reward functions need only be conditioned on the current observation and action. If we try to write down a time-homogeneous Markov transition model for T-Maze, i.e. $T_\omega(\omega' \mid \omega, a)$, we effectively average over all histories consistent with that $(\omega, a)$ pair: $T_\omega(\omega' \mid \omega, a) = \sum_{h \in \mathcal{H}} p_\pi(h \mid \omega, a) p(\omega' \mid \omega, a, h)$. This averages over histories starting with BLUE *or* RED observations, and thus predicts that going UP from the junction will only reach the goal half the time. But the environment does not do any averaging: if the agent initially observed BLUE, going UP from the junction will reach the goal and DOWN will not. In other words, the agent experiences transition dynamics (and, in general, rewards as well) that depend on its complete history and that are inconsistent with the Markov model.

A measure of non-Markovianity must detect when environment dynamics differ from those in the Markov model. One straightforward way to achieve this is to have the agent build two world models—one Markov, one not—and compare them. But, the latter model would require variable-length history inputs, which complicates learning and does not scale with time. Instead, we compare value functions, exploiting the implicit Markov assumption in TD learning and lack thereof in Monte Carlo estimation. This has the advantage that both value functions require only fixed-size inputs, and can be learnt with the same policy.

### 3.1 VALUE FUNCTION ESTIMATION UNDER PARTIAL OBSERVABILITY

Different value function estimators exhibit different behavior in the partially observable setting. In particular, TD($\lambda$) estimators no longer share the same fixed points for all $\lambda$ the way they do in Markov decision processes. Monte Carlo estimation (equivalent to TD($\lambda = 1$)), forms a value estimate by directly averaging samples of the return:

$$\hat{Q}_\pi^{MC}(\omega, a) := \mathbb{E}_\pi[G_t | \omega_t = \omega, a_t = a] \approx \frac{1}{N} \sum_{i=1}^N G_t^{(i)}[\omega_t = \omega, a_t = a],$$

where the expectation is approximated by sampling entire trajectories under the policy. Meanwhile, temporal difference (TD) methods estimate a value function recursively by bootstrapping off of an existing estimate through successive application of the Bellman equation. For TD($\lambda = 0$) this amounts to repeatedly updating $\hat{Q}_\pi^{TD}$ as follows:

$$\hat{Q}_\pi^{TD}(\omega, a) \leftarrow R_t + \gamma \hat{Q}_\pi^{TD}(\omega', a').$$

Modeling the decision process as a POMDP allows us to compute the TD($\lambda$) fixed point in closed form for any parameter $\lambda$ and reactive policy $\pi$:

$$Q_\pi^\lambda = W \Big( I - \gamma T \big( \lambda \Pi^S + (1 - \lambda) \Phi W^\Pi \big) \Big)^{-1} R^{SA}, \tag{4}$$

where $Q_\pi^\lambda$ is an $\Omega \times A$ matrix, and the right hand side is comprised of the following tensors: state weights $W$ ($\Omega \times S$) containing probabilities $\Pr(s|\omega)$ (see Appendix A for the formal definition); identity $I$ (an $S \times A \times S \times A$ tensor with $I_{sas'a'} = \delta_{ss'}\delta_{aa'}$); latent transition dynamics $T$ ($S \times A \times S$); effective policy over latent states $\Pi^S$ ($S \times S \times A$; see Appendix A); observation function $\Phi$ ($S \times \Omega$) containing probabilities $\Pr(\omega|s)$; state-action weights $W^\Pi$ ($\Omega \times S \times A$) containing probabilities $\Pr(s, a|\omega)$; and latent rewards $R^{SA}$ ($S \times A$). **Notation**: Each product between tensors contracts one adjacent index, e.g. for tensors $A$ and $B$, $(AB)_{ijlm} = \sum_k A_{ijk}B_{klm}$, with the exception of the products involving $R^{SA}$ and $T$ on the right hand side, which contract two indices. This derivation is given in Appendix A, and follows the Markov version by Sutton (1988). Setting $\lambda = 0$ or $\lambda = 1$ recovers either the TD or MC value function above, respectively.

Note that the TD estimator replaces $G_t$ with $(R_t + \gamma \hat{Q}_\pi^{TD}(\omega', a'))$, which presupposes that conditioning on $(\omega', a')$ is sufficient for characterizing the distribution of future returns. In other words, TD implicitly makes a Markov assumption. TD averages over all trajectories consistent with the observation-action pair $(\omega', a')$, regardless of whether they are compatible with the preceding experiences, which leads to a biased estimate of value. By contrast, the MC estimator computes expected return using the remainder of the actual, realized trajectory, and is unbiased.[1] This analysis extends to TD($\lambda$) as well, where $\lambda$ controls the strength or weakness of the model's Markov assumption. In practice, it is common to use TD($\lambda$) with a high $\lambda$ in place of MC, since the MC estimator suffers from high variance.

### 3.2 $\lambda$-DISCREPANCY

We have shown that, under partially observability, there may be a discrepancy between $Q_\pi^\lambda$ value functions for two different $\lambda$ parameters due to the implicit Markov assumption in TD($\lambda$). We call this difference the $\lambda$-*discrepancy*, and we will use it as a measure of non-Markovianity.

---

[1]We visualize this distinction for T-Maze in Figure 3 in Appendix E.3.

**Definition 1** *The $\lambda$-discrepancy $\Delta Q_\pi^{\lambda_1,\lambda_2}$ is the weighted $L^2$ norm of the difference between two action-value functions estimated by TD($\lambda$) using different values of $\lambda$:*

$$\Delta Q_{M,\pi}^{\lambda_1,\lambda_2} := \left\| Q_\pi^{\lambda_1} - Q_\pi^{\lambda_2} \right\|_{2,\pi}$$

*where $M$ is the POMDP model, and the norm weighting is as described in Appendix E.2.*

In the remainder of the work, we refer to the $\lambda$-discrepancy as $\Delta Q_\pi^{\lambda_1,\lambda_2}$, i.e. without the subscript $M$, unless otherwise noted. A useful property (that we will prove in Lemma 1) is that the $\lambda$-discrepancy is zero in the fully observable setting. However, for it to be a useful measure of non-Markovianity, we must also show that it is reliably non-zero under partial observability. This leads to the following theorem:

**Theorem 1** *For any POMDP and for any fixed $\lambda$ and $\lambda'$, either $\Delta Q_\pi^{\lambda,\lambda'} \neq 0$ for almost all policies $\pi$ or $\Delta Q_\pi^{\lambda,\lambda'} = 0$ for all policies $\pi$.*

*Proof sketch:* We formulate the $\lambda$-discrepancy as the norm of an analytic function from row-stochastic matrices, the set of matrices whose rows sum to 1, to the reals to show that either the $\lambda$-discrepancy is zero for all policies or is nonzero for almost all policies. The full proof is given in Appendix B.

If the $\lambda$-discrepancy is non-zero for some policy $\pi$, then the $\lambda$-discrepancy is a measure of partial observability and could be used as a learning signal to resolve it. This theorem suggests that almost any stochastic policy in a POMDP will almost always have evidence of partial observability through a non-zero $\lambda$-discrepancy. Even if a POMDP exhibits a zero $\lambda$-discrepancy for a particular policy, this theorem further suggests a way to avoid this case: small perturbations in the policy (for instance, with an $\epsilon$-greedy policy).

We now consider when the $\lambda$-discrepancy is 0 for all policies, and show that these POMDPs are either Markov or uninteresting.

### 3.3 WHEN IS THE $\lambda$-DISCREPANCY ZERO?

Because norms are positive definite, to analyze the case in Theorem 1 where the $\lambda$-discrepancy is 0 for all policies, it suffices to consider the expression inside the norm of Definition 1:

$$W\left(A_\pi^{\lambda_1} - A_\pi^{\lambda_2}\right)R^{SA}, \quad \text{where } A_\pi^\lambda = \left(I - \gamma T\left(\lambda\Pi^S + (1-\lambda)\Phi W^\Pi\right)\right)^{-1}. \tag{5}$$

The only ways for this equation to be zero are when the difference term $A_\pi^{\lambda_1} - A_\pi^{\lambda_2}$ is zero (which we will show to be an MDP), or in the special case where this difference term is projected away by the outer terms $W$ and/or $R^{SA}$. We first consider when the two inner terms—which are the only terms that depend on $\lambda$—are equal, i.e.:

$$A_\pi^{\lambda_1} = A_\pi^{\lambda_2}. \tag{6}$$

In this case the system is a *block MDP*, a POMDP / MDP hybrid model featuring Markov observations, each of which can only be produced by one unique latent state (Du et al., 2019). Block MDPs are thus Markov over both states and observations.

**Lemma 1** *For any POMDP and any $\lambda, \lambda'$, Equation 6 holds if and only if the system is a block MDP.*

*Proof:* See Appendix C.

Now we consider POMDPs where the difference between $A_\pi^\lambda$ is projected away by the outer terms $W$ and $R^{SA}$. To see how a $\lambda$-discrepancy of 0 is possible in this case, we first expand Equation 4 as a power series,

$$Q_\pi^\lambda = WR^{SA} + \gamma WT\left(\lambda\Pi^S + (1-\lambda)\Phi W^\Pi\right)R^{SA} +$$
$$\gamma^2 WT\left(\lambda\Pi^S + (1-\lambda)\Phi W^\Pi\right)T\left(\lambda\Pi^S + (1-\lambda)\Phi W^\Pi\right)R^{SA} + \ldots,$$

and consider when $Q_\pi^{\lambda_1} - Q_\pi^{\lambda_2}$ is zero. This could occur due to cancellation of the terms between the two power series. One case of this cancellation occurs when $\lambda = 0$ and $\lambda' = 1$. We can group the

power series expansion by $\gamma^n$ coefficients:

$$Q_\pi^0 - Q_\pi^1 = \gamma \left( WT\Pi^S R^{SA} - WT\Phi W^\Pi R^{SA} \right)$$
$$+ \gamma^2 \left( WT\Pi^S T\Pi^S R^{SA} - WT\Phi W^\Pi T\Phi W^\Pi R^{SA} \right) + \dots$$

Setting each grouped term to 0, this gives a class of POMDPs with 0 $\lambda$-discrepancy, namely those where the *expected* reward at each time step equals the expected rewards given Markovian observation rollouts. Furthermore, this characterization of the $\lambda$-discrepancy actually holds for almost all $\gamma$:

**Lemma 2** *For any POMDP, $\Delta Q_\pi^{0,1} = 0$ if for all initial observations $o^0$ and actions $a^0$, and all horizons $T$,*

$$\mathbb{E}(r^T|\omega^0, a^0) = \sum_{\omega^1, \dots, \omega^T} \mathbb{E}(r^T|\omega^T)\mathbb{P}(\omega^1|\omega^0, a^0) \prod_{t=1}^{T-1} \mathbb{P}(\omega^{t+1}|\omega^t).$$

*Furthermore, the converse holds for almost all choices of the discount factor $\gamma$.*

*Proof:* See Appendix D.

In terms of implementation, this implies that if $\gamma$ is chosen uniformly from some interval in $(0, 1)$, there is zero probability that the $\lambda$-discrepancy is 0 unless it is 0 for all $\gamma$.

Through Theorem 1 and analyzing redundant or unlikely cases where $\lambda$-discrepancy is 0, we conclude that the $\lambda$-discrepancy is very likely to be a useful measure for detecting and mitigating partial observability. Theorem 1 also implies that if $\lambda$-discrepancy is non-zero, zeroing it with some memory function will result in memory-augmented observations that are Markov. It is also a promisingly practical measure, since it relies on the very value functions that form the foundation of most RL methods and for which many well-developed learning algorithms exist. In the next section, we demonstrate the efficacy of using the $\lambda$-discrepancy to learn memory functions that reduce partial observability in well-known POMDPs.

## 4   MEMORY LEARNING WITH THE $\lambda$-DISCREPANCY

Since single observations are insufficient for optimal decision making, the agent must augment its observations with a summary of its past. We formalize this concept as a *memory function*. If $\mathcal{H}$ is the space of all possible histories, a *memory function* $\mu : \mathcal{H} \mapsto m$ is a mapping from a variable-length history $h_t \doteq (\omega_0, a_0, r_0, \dots, \omega_{t-1}, a_{t-1}, r_{t-1}, \omega_t) \in \mathcal{H}$ to a memory state $m$ within a set of possible memory states $M$. This definition captures the broadest class of memory functions, but for practical reasons, we will restrict our focus to *recurrent* memory functions that use fixed-size inputs and update their memory state incrementally. A recurrent memory function $\mu : O \times A \times M \to M$ takes in a previous memory state $m$ (as well as previous observation $\omega$ and action $a$), and outputs a next memory state $m' = \mu(\omega, a, m)$. For example, in the T-Maze environment of Figure 1, one useful memory function modifies the memory state based on the initial observation (BLUE or RED), then keeps the memory state fixed until the purple junction. Memory functions naturally lead to memory-augmented policies $\pi_\mu : (\Omega \times M) \to A$ and value functions $V_{\pi,\mu} : (\Omega \times M) \to \mathbb{R}$ and $Q_{\pi,\mu} : (\Omega \times M) \times A \to \mathbb{R}$ that reflect the expected return under such policies, all expressed in terms of the newly defined observation space $(\Omega \times M)$.

### 4.1   $\lambda$-DISCREPANCY FOR MEMORY FUNCTIONS

As currently defined, the $\lambda$-discrepancy can identify when we need memory, but it cannot tell us what to remember. We therefore replace the observation value functions in Definition 1 with their memory-augmented counterparts:

$$\Delta Q_{\pi,\mu}^{\lambda_1,\lambda_2} := \Delta Q_{M^\mu,\pi}^{\lambda_1,\lambda_2} \tag{7}$$

where $M^\mu$ augments POMDP $M$ with memory function $\mu$ as in Appendix E.1, and the right hand side is the $\lambda$-discrepancy defined with respect to a POMDP $M^\mu$ as given in Definition 1. The augmented observations $\Omega \times M$ constitute a Markov state if the memory is a sufficient statistic of history. In this section, we set $\lambda_1 = 0$ and $\lambda_2 = 1$, which represent the TD(0) and MC value functions respectively.

---

**Algorithm 1** Memory Optimization with Value Improvement

---

**Input:** Randomly initialized policy parameters $\theta_\pi$, where $\Pi = \text{softmax}(\theta_\pi)$, randomly initialized memory parameters $\theta_\mu$, POMDP parameters $\mathcal{P} := (T, R^{SA}, \phi, p_0, \gamma)$, number of memory improvement steps $n_{\text{steps},M}$, number of policy iteration steps $n_{\text{steps},\pi}$, learning rate $\alpha \in [0,1]$, and number of initial random policies $n$.
*// Calculate memoryless optimal policy.*
$\theta_{\pi*} \leftarrow \texttt{policy\_improvement}(\theta_\pi, \mathcal{P}, n_{\text{steps},\pi})$
*// Initialize random policies and select policy to fix for memory learning.*
$\{\theta_0, \ldots, \theta_{n-1}\} \leftarrow \texttt{randomly\_init\_n\_policies}(n)$
$\theta_\pi \leftarrow \texttt{select\_argmax\_lambda\_discrepancy}(\{\theta_{\pi*}, \theta_0, \ldots, \theta_{n-1}\})$
*// Repeat policy over all memory states.*
$\theta_{\pi_\mu} \leftarrow \texttt{repeat}(\theta_\pi, |M|)$
*// Improve memory function.*
$\theta_\mu \leftarrow \texttt{memory\_improvement}(\theta_\mu, \theta_{\pi_\mu}, \mathcal{P}, n_{\text{steps},M})$
*// Improve memory-augmented policy over learnt-memory-augmented POMDP*
$\theta_{\pi_\mu} \leftarrow \texttt{policy\_improvement}(\theta_{\pi_\mu}, \mathcal{P}_{\theta_\mu}, n_{\text{steps},\pi})$
**return** $\theta_{\pi_\mu}, \theta_\mu$

---

We test two approaches for learning a memory function by minimizing $\lambda$-discrepancy. We begin with a gradient-based optimization algorithm for memory optimization in the planning setting, where the algorithm has access to the ground-truth POMDP dynamics (transition dynamics $T$, reward $R^{SA}$, observation function $\Phi$) and can calculate gradients through these quantities. Next, we relax this assumption and consider a discrete optimization approach that does not necessarily require the ground-truth POMDP dynamics for updates. It descends the $\lambda$-discrepancy with a hill-climbing algorithm.

We show that under both optimization schemes, gradient-based and hill-climbing, minimizing the $\lambda$-discrepancy improves performance over the baseline memoryless policy in a collection of classic POMDP problems. Together, these results demonstrate the value of $\lambda$-discrepancy as a measure of partial observability, and for solving general decision processes.

## 4.2 MEMORY OPTIMIZATION WITH GRADIENTS

Our first algorithm calculates and optimizes closed-form gradients of the $\lambda$-discrepancy with respect to a parameterized memory function. We calculate memory-augmented value functions as in Equation 4. This requires generalizing our definition of memory functions to *stochastic* memory functions $\mu : O \times A \times M \times M \rightarrow [0,1]$, so that the optimization surface is continuous and gradients exist. Stochastic memory functions are transition functions between memory states that depend on observables of the POMDP. At every step, the memory function takes as input an observation, action, and previous memory state, and stochastically outputs the memory state at the next time step. We then augment the POMDP dynamics and policy by combining the memory function with the state transition function using a *memory Cartesian product* operation. Full details of this memory augmentation procedure are in Appendix E.1, and the practical implementation is in E.4.

Algorithm 1 describes our memory optimization procedure, which reduces $\lambda$-discrepancy to learn a memory function, then learns an optimal memory-augmented policy. The `memory_improvement` subroutine can be either gradient-based (this section) or hill-climbing (next section). Further algorithm details are in Appendix E.4.

**Experiments.** We conduct experiments on a range of classic partially observable decision problems requiring skills like memorization, uncertainty estimation, and counting. The environments are: T-Maze (Bakker, 2001), the Tiger problem (Cassandra et al., 1994), Paint (Kushmerick et al., 1995), Cheese Maze, Network, Shuttle (Chrisman, 1992), and the $4 \times 3$ maze (Russell and Norvig, 1995). In Figure 2, we compare normalized learning performance between memoryless (blue bars) and $k$-bit memory-augmented agents (respectively yellow, orange, and red solid bars, for $k \in \{1, 2, 3\}$).[2] Performance is expected return, normalized to the range between that of a random policy and the optimal belief-state policy learnt from a POMDP solver (Cassandra, 2003). Policy improvement uses

---

[2]Note that $k$ memory bits correspond to $2^k$ memory states.

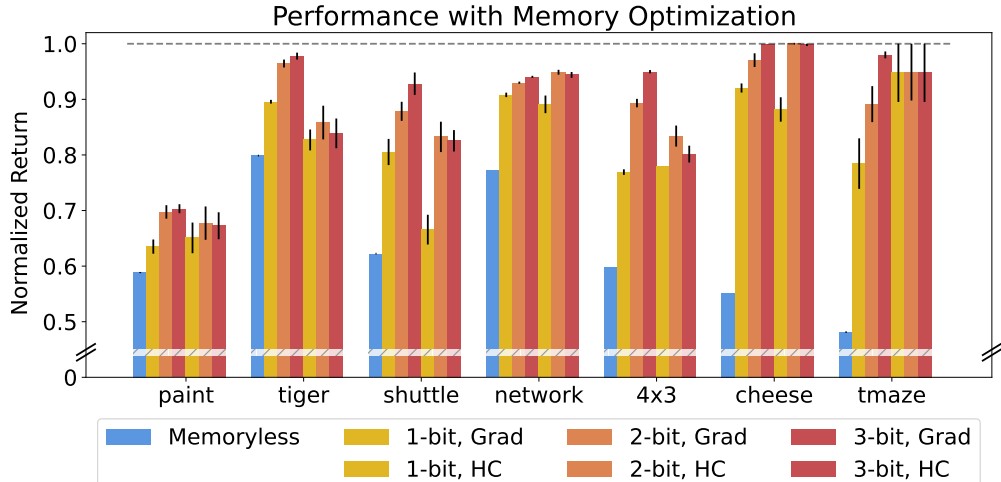

Figure 2: Memory optimization increases normalized return of subsequent policy gradient learning. Solid bars denote gradient-based memory optimization (Grad) and hatched bars denote hill climbing (HC). Performance is calculated as the average start-state value weighted by the start state distribution, and is normalized between a random policy ($y = 0$) and the optimal belief state policy ($y = 1$) found with a POMDP solver (Cassandra, 2003). Error bars are standard error of the mean over 30 seeds.

the policy gradient (Sutton et al., 2000) algorithm, since it one of the most effective ways to optimize stochastic policies in partially observable environments (Sutton and Barto, 2018).[3] Full experimental details are provided in Appendix E.3.

Even with just a single bit of memory, we see large performance gains for all but two of the problems we test on. As the number of memory bits increases, so does the performance. This suggests that the $\lambda$-discrepancy is a useful objective for learning memory functions to resolve many forms of partial observability. Additionally, the learnt memory functions are often quite sensible. Figure 1b shows a visualization of one such memory function learnt for T-Maze. Despite optimizing over stochastic memories, the function is essentially fully deterministic: it sets the memory state according to the initial observation, and holds it constant otherwise. This allows the agent to concisely express the optimal policy in terms of augmented observations.

### 4.3 MEMORY OPTIMIZATION WITH HILL CLIMBING

Gradient-based optimization is effective at learning memory functions, but requires differentiating—in closed form—through the quantities $\Phi$, $T$, and $R$, which in principle cannot be observed by the agent. In this section, we consider a more realistic optimization procedure that requires only point-estimates of the relevant value functions. We employ a hill climbing algorithm based on simulated annealing (Kirkpatrick et al., 1983) that searches the space of deterministic memory functions to minimize the $\lambda$-discrepancy using these point estimates.

We replace the `memory_improvement` subroutine of Algorithm 1 with simulated annealing, and define a local search "neighborhood" over discrete memory functions so the algorithm can propose successors. We run simulated annealing for a fixed number of steps, with random restarts and hyperparameter resampling to improve robustness. See Appendix E.6 for more details.

For simplicity, and to isolate the effects of memory learning with the $\lambda$-discrepancy, we assume the agent has access to modules for accurate value estimation and effective policy improvement. These modules could easily be replaced with sampled versions, but using a closed-form implementation allows us to test our scientific claims about memory optimization without introducing confounding variables. Here we only seek to show the viability of the $\lambda$-discrepancy as a measure of partial observability and as a training signal for learning memory. We leave an exploration of its interaction with value and policy learning algorithms for future work.

---

[3]See Appendix E.3.4 for a version that uses policy iteration (Howard, 1960) instead.

**Experiments.** We repeat the experiments of Section 4.2, this time using hill climbing, and plot the results side by side with the original results in Figure 2 as hashed bars. Across all environments, the hill-climbing optimization procedure led to similar levels of final performance as the "best-case" gradient-based procedure, in some cases even matching the performance of the optimal belief-state policy. This suggests that in addition to being theoretically interesting, the $\lambda$-discrepancy has practical value for learning memory in general decision processes.

## 5 RELATED WORK

Algorithms to resolve partial observability for non-Markov decision processes have been studied extensively. The most popular class of solution methods for POMDPs are *belief-state* methods, which have seen substantial success in robotics (Thrun et al., 2005). However, belief-state approaches are intractable for even small environments where the hidden state space or its dynamics are unknown (Zhang et al., 2012). Predictive state representations (Littman et al., 2001) resolve partial observability with *tests* about the future trajectory. These tests are difficult to construct and only works on relatively small domains (Zhang et al., 2012). Another approach is to leverage general value functions (Sutton et al., 2011) (GVFs) to make predictions about the environment that can encompass history information. GVFs have been shown to help in select non-Markov decision processes (Schlegel et al., 2021), but discovering the right GVFs for each problem (Veeriah et al., 2019) and their applicability for history summarization are open, unanswered questions.

Memory-based, history-summarization approaches to solving POMDPs have also been considered extensively. Meuleau et al. learnt memory via stochastic gradient descent in the parameters of a policy graph that encoded actions as nodes and observations as edges (Meuleau et al., 1999). Baxter and Bartlett's GPOMDP algorithm used a biased gradient estimate to enable policy gradients to work for POMDPs; GPOMDP can be extended to work with finite sequences of observations or parameterized belief states as well (Baxter and Bartlett, 2001). Hansen developed policy iteration and heuristic search finite-state controller methods that improved on previous value iteration-based approaches. Each node in his controller corresponded to a portion of the value function that then specified a greedy action choice (Hansen, 1998). McCallum developed algorithms based on the "utile distinction test" that determines when two histories should be considered usefully distinct and the "nearest sequence memory" principle, which suggests past experiences should be considered in estimating value (McCallum, 1996). The methods of Meuleau, Baxter, and Hansen did not make a distinction between memory function parameters and policy parameters as we do in this work. MaCallum's work did consider a role for memory more similar to our work, but while his memory states could only capture a finite length of past observation history, our memory functions can enable the agent to remember for an arbitrary length of time.

Finally, most modern approaches use recurrent neural networks (RNNs) trained via backpropagation through time (BPTT) to tackle non-Markov decision processes. While this method works with many environments (Lin and Mitchell, 1993; Bakker, 2001; Ni et al., 2021), their success is sensitive to architecture and hyperparameter choices (Ni et al., 2021), and requires that both value and memory be learnt simultaneously. Overall, there is a dearth of scalable solution methods that specifically tackles partial observability.

## 6 CONCLUSION

We introduce the $\lambda$-discrepancy: an observable and minimizable measure of non-Markovianity for resolving partial observability. The $\lambda$-discrepancy is the norm between two TD($\lambda$) fixed-points for two different values of $\lambda$s. We motivate and prove several theoretical properties that make the $\lambda$-discrepancy a reasonable optimization objective for learning effective recurrent memory functions. We empirically test the efficacy of this measure for solving general, non-Markov decision processes through experiments on a broad range of partially observable environments in two problem settings: the gradient-based planning setting, and the online sample-based setting. We find that reducing the $\lambda$-discrepancy leads to memory functions that can solve non-Markov decision processes, and scale with the size of the decision process. We conclude that the $\lambda$-discrepancy is a robust measure for improving decision making in non-Markov decision processes.

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

## A TD($\lambda$) FIXED POINT

Here we derive the fixed point of the TD($\lambda$) action-value update rule in a POMDP. First, define the expected return given initial observation $\omega_0$ and initial action $a_0$ as

$$
\begin{aligned}
\mathbb{E}_\Pi(G^n|\omega_0,a_0) = &\sum_{s_0}\mathbb{P}(s_0|\omega_0)\sum_{s_1}\mathbb{P}(s_1|s_0,a_0)\sum_r\mathbb{P}(r_0|s_0,a_0,s_1)r_0 \\
&+ \gamma\sum_{s_0}\mathbb{P}(s_0|\omega_0)\sum_{s_1}\mathbb{P}(s_1|s_0,a_0)\sum_{\omega_1}\sum_{a_1}\sum_{s_2}\mathbb{P}(\omega_1|s_1)\mathbb{P}(a_1|\omega_1)\mathbb{P}(s_2|s_1,a_1) \\
&\qquad\qquad\qquad\qquad\qquad\qquad\qquad\qquad\qquad\sum_{r_1}\mathbb{P}(r_1|s_1,a_1,s_2)r_1 \\
&+ \gamma^2\sum_{s_0}\mathbb{P}(s_0|\omega_0)\sum_{s_1}\mathbb{P}(s_1|s_0,a_0)\sum_{\omega_1}\sum_{a_1}\sum_{s_2}\mathbb{P}(\omega_1|s_1)\mathbb{P}(a_1|\omega_1)\mathbb{P}(s_2|s_1,a_1) \\
&\qquad * \sum_{\omega_2}\sum_{a_2}\sum_{s_3}\mathbb{P}(\omega_2|s_2)\mathbb{P}(a_2|\omega_2)\mathbb{P}(s_3|s_2,a_2)\sum_{r_2}\mathbb{P}(r_2|s_2,a_2,s_3)r_2 \\
&+ \dots
\end{aligned}
$$

We can define the $n$-step bootstrapped update rule from this given a value matrix $Q$ by replacing part of the term with coefficient $\gamma^n$ with a $Q$ value, e.g. for the $n=2$ case, we get

$$
\begin{aligned}
Q(\omega_0,a_0) \leftarrow &\sum_{s_0}\mathbb{P}(s_0|\omega_0)\sum_{s_1}\mathbb{P}(s_1|s_0,a_0)\sum_r\mathbb{P}(r_0|s_0,a_0,s_1)r_0 \\
&+ \gamma\sum_{s_0}\mathbb{P}(s_0|\omega_0)\sum_{s_1}\mathbb{P}(s_1|s_0,a_0)\sum_{\omega_1}\sum_{a_1}\sum_{s_2}\mathbb{P}(\omega_1|s_1)\mathbb{P}(a_1|\omega_1)\mathbb{P}(s_2|s_1,a_1) \\
&\qquad\qquad\qquad\qquad\qquad\qquad\qquad\qquad\qquad\sum_{r_1}\mathbb{P}(r_1|s_1,a_1,s_2)r_1 \\
&+ \gamma^2\sum_{s_0}\mathbb{P}(s_0|\omega_0)\sum_{s_1}\mathbb{P}(s_1|s_0,a_0)\sum_{\omega_1}\sum_{a_1}\sum_{s_2}\mathbb{P}(\omega_1|s_1)\mathbb{P}(a_1|\omega_1)\mathbb{P}(s_2|s_1,a_1) \\
&\qquad * \sum_{\omega_2}\mathbb{P}(\omega_2|s_2)\sum_{a_2}\mathbb{P}(a_2|\omega_2)Q(\omega_2,a_2)
\end{aligned}
$$

Translating these expressions into matrix notation, we have

$$
\begin{aligned}
\mathbb{E}_\Pi(G^n|\omega_0,a_0) = &\sum_{s_0}W_{\omega_0,s_0}\sum_{s_1}T_{s_0,a_0,s_1}R_{s_0,a_0} \\
&+ \gamma\sum_{s_0}W_{\omega_0,s_0}\sum_{s_1}T_{s_0,a_0,s_1}\sum_{\omega_1}\sum_{a_1}\sum_{s_2}\Phi_{s_1,\omega_1}\pi_{\omega_1,a_1}T_{s_1,a_1,s_2}R_{s_1,a_1} \\
&+ \gamma^2\sum_{s_0}W_{\omega_0,s_0}\sum_{s_1}T_{s_0,a_0,s_1}\sum_{\omega_1}\sum_{a_1}\sum_{s_2}\Phi_{s_1,\omega_1}\pi_{\omega_1,a_1}T_{s_1,a_1,s_2} \\
&* \sum_{\omega_2}\sum_{a_2}\sum_{s_3}\Phi_{s_2,\omega_2}\pi_{\omega_2,a_2}T_{s_2,a_2,s_3}R_{s_2,a_2} \\
&+ \dots
\end{aligned}
$$

where the terms $W$, $T$, and $R$, are as in Equation 4, and $\pi$ is the $\Omega \times A$ policy. In particular, $W_{\omega,s} = \mathbb{P}(s|\omega)$, which averages $\mathbb{P}(s_t|\omega_t)$ over all timesteps, weighted by visitation probability and discounted by $\gamma$. This is a well-defined stationary quantity, and it can be computed as follows. First solve the system $Ax = b$ to find the discounted state occupancy counts $x = c(s)$, where $A = (I - \gamma(T^\pi)^\top)$ accounts for the policy-dependent state-to-state transition dynamics $T^\pi$, and $b = p_0$ is the initial state distribution over $s$. Then $\mathbb{P}(s|\omega) \propto c(s) * \mathbb{P}(\omega|s)$, so we can just multiply these terms together and renormalize. For the bootstrapped update rule, we have

$$
\begin{aligned}
Q(\omega_0,a_0) \leftarrow &\sum_{s_0}W_{\omega_0,s_0}\sum_{s_1}T_{s_0,a_0,s_1}R_{s_0,a_0} \\
&+ \gamma\sum_{s_0}W_{\omega_0,s_0}\sum_{s_1}T_{s_0,a_0,s_1}\sum_{\omega_1}\sum_{a_1}\sum_{s_2}\Phi_{s_1,\omega_1}\pi_{\omega_1,a_1}T_{s_1,a_1,s_2}R_{s_1,a_1} \\
&+ \gamma^2\sum_{s_0}W_{\omega_0,s_0}\sum_{s_1}T_{s_0,a_0,s_1}\sum_{\omega_1}\sum_{a_1}\sum_{s_2}\Phi_{s_1,\omega_1}\pi_{\omega_1,a_1}T_{s_1,a_1,s_2} \\
&* \sum_{\omega_2}\Phi_{s_2,\omega_2}\sum_{a_2}\Pi_{\omega_2,a_2}Q(\omega_2,a_2)
\end{aligned}
$$

Therefore, the $n$-step update rule given a matrix of action-values $Q$ is

$$
Q \leftarrow Q_n(Q) := W\left(\sum_{k=0}^{n-1}(\gamma T\Pi^S)^k R^{SA} + \gamma(\gamma T\Pi^S)^{n-1}T\Phi\Pi Q\right)
$$

where $\Pi$ is an $\Omega \times \Omega \times A$ representation of the $\Omega \times A$ policy $\pi$ with $\Pi_{\omega,\omega',a} = \delta_{\omega,\omega'}\pi_{\omega,a}$, $\Pi^S$, the effective policy over latent states, is likewise an $S \times S \times A$ representation of the matrix $\Phi \cdot \pi$, and $R^{SA}_{s,a} = \sum_{s'} T_{s,a,s'} R^{SAS}_{s,a,s'}$.

**Notation**: We use non-standard notation here for tensor contractions to avoid being overly verbose. Unless otherwise noted, all contractions contract 1 index except those involving $R^{SA}$ and $T$ on the right hand side and those with $W^A$ on the left hand side, which contract 2 indices.

We also have the standard definition of the TD($\lambda$) update rule as $Q \leftarrow (1-\lambda)\sum_{n=1}^{\infty} \lambda^{n-1} Q_n(Q)$. We are concerned with the fixed point of this update rule, which we refer to as $Q^\lambda$:

$$Q^\lambda = (1-\lambda)\sum_{n=1}^{\infty} \lambda^{n-1} W \left( \sum_{k=0}^{n-1} (\gamma T\Pi^S)^k R^{SA} + \gamma(\gamma T\Pi^S)^{n-1} T\Phi\Pi Q^\lambda \right).$$

Separating this into a reward part with factor $R^{SA}$ and a value part with factor $Q^\lambda$, we find that the value part is

$$(1-\lambda)W\left(\sum_{n=1}^{\infty}(\lambda\gamma T\Pi^S)^{n-1}\right)\gamma T\Phi\Pi Q^\lambda$$
$$= (1-\lambda)W(I-\lambda\gamma T\Pi^S)^{-1}\gamma T\Phi\Pi Q^\lambda,$$

and for the reward part, we have the coefficients of $R^{SA}$ in the table below for values of $n$ and $k$

|        | $k=0$       | 1                      | 2                         | $\ldots$ |
|--------|-------------|------------------------|---------------------------|----------|
| $n=1$  | 1           |                        |                           | $\ldots$ |
| 2      | $\lambda$   | $\lambda\gamma T\Pi^S$ |                           | $\ldots$ |
| 3      | $\lambda^2$ | $\lambda^2\gamma T\Pi^S$ | $\lambda^2(\gamma T\Pi^S)^2$ | $\ldots$ |
| $\vdots$ | $\vdots$  | $\vdots$               | $\vdots$                  | $\ddots$ |

where each term is multiplied by $(1-\lambda)W$ in front. We can then see by summing over rows before columns that the reward part is:

$$(1-\lambda)W\sum_{k=0}^{\infty}\frac{1}{1-\lambda}(\lambda\gamma T\Pi^S)^k R^{SA}$$
$$= W(I-\lambda\gamma T\Pi^S)^{-1}R^{SA}.$$

So we rewrite $Q^\lambda$ as follows:

$$Q^\lambda = W\left((I-\lambda\gamma T\Pi^S)^{-1}\left(R^{SA}+(1-\lambda)\gamma T\Phi\Pi Q^\lambda\right)\right).$$

Now let $W^A = W \otimes I_{A,A}$ and $\left(W^\Pi\right)_{ijk} = \Pi W^A$. Here, $\otimes$ means the Kronecker product. This essentially repeats the $W$ matrix $A$ times to incorporate actions into the tensor. Note that for any $S \times A$ tensor $G$, $W^A G = WG$. This is because $(W^A G)_{ij} = \sum_{k,l} W^A_{ijkl} G_{kl}$, and the only nonzero terms in this sum are those such that $j = l$. For these indices, $W^A_{ijkl} = W_{ik}$, so $\sum_{k,l} W^A_{ijkl} G_{kl} = \sum_k W_{ik} G_{kj} = (WG)_{ij}$.

Also, let $F = \left(I - \lambda\gamma T\Pi^S\right)^{-1}$. Then we find:

$$Q^\lambda = W^A\left(\left(I-\lambda\gamma T\Pi^S\right)^{-1}\left(R^{SA}+(1-\lambda)\gamma T\Phi\Pi Q^\lambda\right)\right)$$
$$= W^A\left(F\left(R^{SA}+(1-\lambda)\gamma T\Phi\Pi Q^\lambda\right)\right)$$
$$= W^A F R^{SA} + W^A F(1-\lambda)\gamma T\Phi\Pi Q^\lambda$$

At which point we can subtract the second term on the right hand side from both sides, factor out $Q^\lambda$ on the right, and multiply by $\left(I - (1-\lambda)\gamma W^A F T\Phi\Pi\right)^{-1}$ on the left of both sides to obtain:

$$Q^\lambda = \left(I - (1-\lambda)\gamma W^A F T\Phi\Pi\right)^{-1} W^A F R^{SA}$$
$$= W^A\left(I - (1-\lambda)\gamma F T\Phi\Pi W^A\right)^{-1} F R^{SA}$$
$$= W\left(I - (1-\lambda)\gamma F T\Phi W^\Pi\right)^{-1} F R^{SA}$$
$$= W\left(F + (1-\lambda)\gamma F T\Phi W^\Pi F + \ldots + (1-\lambda)^k\gamma^k F T\Phi W^\Pi F T\Phi W^\Pi F + \ldots\right) R^{SA},$$

where the last equality follows from expanding the geometric series. Now we use the identity $(A - B)^{-1} = \sum_{k=0}^{\infty} (A^{-1}B)^k A^{-1}$ to find:

$$Q^\lambda = W \left( F^{-1} - (1 - \lambda)\gamma T\Phi W^\Pi \right)^{-1} R^{SA}$$
$$= W \left( I - \gamma T \left( \lambda\Pi^S + (1 - \lambda)\Phi W^\Pi \right) \right)^{-1} R^{SA}.$$

To recap our previous definitions, $W$ is an $\Omega \times S$ tensor, $I$ is an $S \times A \times S \times A$ tensor, $T$ is an $S \times A \times S$ tensor, $\Pi^S$ is an $S \times S \times A$ tensor, $\Phi$ is an $S \times \Omega$ tensor, $W^\Pi$ is an $\Omega \times S \times A$ tensor, and $R^{SA}$ is an $S \times A$ tensor. Thus, the operation between $W$ and the tensor inverse is a single summation over the last index of $W$ and the first index of the inverse. Operations within the matrix inverse are likewise tensor dot products (single index contractions), and the operation between the tensor inverse and $R^{SA}$ is a tensor double contraction.

Lastly, we briefly note that one can get the $V$ values by replacing $W$ in the above equation with $W^\Pi$ and changing the operation involving it on the right from a dot product to a double contraction. We can confirm that $V_o = \sum_a \pi_{o,a} Q_{o,a}$, by rewriting the expression on the right as follows:

$$\sum_a \pi_{o,a} Q_{o,a} = \sum_a \pi_{o,a} \sum_{s,a'} W^A_{o,a,s,a'} B_{s,a'}$$
$$= \sum_a \pi_{o,a} \sum_s W^A_{o,a,s,a} B_{s,a}$$
$$= \sum_a \pi_{o,a} \sum_s W_{o,s} B_{s,a}$$
$$= \sum_{s,a} \underbrace{\pi_{o,a} W_{o,s}}_{W^\Pi_{o,s,a}} B_{s,a}$$
$$= V_o,$$

where $B = \left( I - \gamma T \left( \lambda\Pi^S + (1 - \lambda)\Phi W^\Pi \right) \right)^{-1} R^{SA}$ is an $S \times A$ tensor.

## B    PROOF OF THEOREM 1 (ALMOST ALL)

In this section we prove Theorem 1, that there is either a $\lambda$-discrepancy for almost all policies or for no policies. Fix $\lambda$ and $\lambda'$. Recall that we define the $\lambda$-discrepancy as follows:

$$\Delta Q_\pi := \left\| Q_\pi^\lambda - Q_\pi^{\lambda'} \right\|_{2,\pi} = \left\| \left( W^\Pi \left( A_\pi^\lambda - A_\pi^{\lambda'} \right) R^{SA} \right) \cdot w_\pi \right\|_2$$

where $A_\pi^\lambda = \left( I - \gamma T \left( \lambda \Pi^S + (1-\lambda)\Phi W^\Pi \right) \right)^{-1}$ and $w_\pi$ is a weight vector of probabilities dependent on $\pi$ defined as $w_\pi(\omega, a) = (1, \pi(a|\omega))$. Let $U$ be the largest open set in the space of stochastic $\Omega \times A$ matrices, considered as a subset of $\mathbb{R}^{\Omega(A-1)}$. Now consider the lambda discrepancy as a function of the policy $\pi$. In other words, we define

$$\Delta Q : U \to \mathbb{R}$$

$$\pi \mapsto Q_\pi^\lambda - Q_\pi^{\lambda'}$$

Let $U$ be an open subset of $\mathbb{R}^n$. We say that a function $f : U \to \mathbb{R}$ is real analytic on $U$ if for all $x \in U$, $f$ can be written as a convergent power series in some neighborhood of $x$. For this proof, we will utilize the following facts: 1) the composition of analytic functions is analytic (Krantz and Parks, 2002), 2) the quotient of two analytic functions is analytic where the denominator is nonzero, 3) a real analytic function on a domain $U$ is either identically $0$ or only zero on a set of measure $0$ (Mityagin, 2020).

We will also use the fact that for $A$ an invertible matrix, each entry $A_{ij}^{-1}$ is analytic in the entries of $A$ where the entries of $A$ yield a nonzero determinant. We can prove this by first writing $A^{-1} = \det(A)^{-1} \operatorname{adj}(A) = \det(A)^{-1} C^T$ where $\operatorname{adj} A$ is the adjugate of $A$ and $C$ is the cofactor matrix of $A$. Each entry of the cofactor matrix is a cofactor that is polynomial in the entries of $A$, and is therefore analytic in them. Therefore, each entry of $A^{-1}$ is the quotient of two analytic functions and is therefore analytic except where $\det A = 0$.

Next, we will show that $\Delta Q$ is an analytic function. Note that the variable terms in the equation are $W$, $W^\Pi$, $R^{SA}$, $T$, $\Pi^S$, and $\Phi$. $R^{SA}$, $T$, and $\Phi$ are constant with respect to $\pi$. $\Pi_{ilj}^S = \sum_k \delta_{il} \Phi_{ik} \pi_{kj}$, so each entry of $\Pi^S$ is analytic on $U$ in the entries of $\pi$. Likewise, $P_{ij} = \sum_{k,a} \Phi_{ik} \pi_{ka} T_{iaj}$ is analytic on $U$. Therefore, the state-occupancy counts $c = \mu_0 + \gamma P^T \mu_0 + \gamma^2 (P^T)^2 \mu_0 + \cdots = (I - \gamma P^T)^{-1} \mu_0$ are the composition of analytic functions and thus analytic on $U$. $W_{ij} = \frac{\Phi_{ji} c_j}{\sum_k \Phi_{ki} c_k}$ is analytic on $U$ for the same reason, and the denominator of $W_{ij}$, $\sum_k \Phi_{ki} c_k$, is nonzero for all observations able to be observed with nonzero probability. Finally, $\Delta Q$ is then a composition of analytic functions on $U$ and thus analytic itself.

To handle the norm weighting, we note that $w_\pi$ is analytic in $\pi$ as $w_\pi = (1, \pi(a|\omega))$, and the dot product of $w_\pi$ with $\Delta Q$ is also analytic. Now, we use the fact mentioned above that the zero set of a nontrivial analytic function is of measure zero. Therefore, the zero set of $\Delta Q \cdot w_\pi$ is either zero for all policies or zero only on a set of measure zero. To finish, we note that because norms are positive definite, $\Delta Q_\pi = 0$ if and only if $\Delta Q \cdot w_\pi = 0$, so this result extends to the normed $\lambda$-discrepancy as well.

## C    Proof of Lemma 1 (Block MDP)

In this section, we prove Lemma 1 concerning when the system is a Block MDP. Recall that in Eq. equation 5 we define $A_\pi^\lambda = \left(I - \gamma T \left(\lambda \Pi^S + (1 - \lambda)\Phi W^\Pi\right)\right)^{-1}$. Suppose $A_\pi^\lambda = A_\pi^{\lambda'}$. Then $\gamma T \left(\lambda \Pi^S + (1 - \lambda)\Phi W^\Pi\right) = \gamma T \left(\lambda' \Pi^S + (1 - \lambda')\Phi W^\Pi\right)$ as matrix inverses are unique. We can rewrite this as $(\lambda - \lambda')\Pi^S - (\lambda - \lambda')\Phi W^\Pi = (\lambda - \lambda')(\Pi^S - \Phi W^\Pi) = 0$. This implies that either $\lambda = \lambda'$ or $\Pi^S = \Phi W^\Pi$.

Writing $\Pi^S = \Phi W^\Pi$ out in terms of probability, this implies that for all $i, j, k$, $\sum_\omega \mathbb{P}(\omega|s_i)\mathbb{P}(a_k|\omega) = \sum_\omega \mathbb{P}(\omega|s_i)\mathbb{P}(a_k|\omega)\mathbb{P}(s_j|\omega)$ if $i = j$, and $\sum_\omega \mathbb{P}(\omega|s_i)\mathbb{P}(a_k|\omega)\mathbb{P}(s_j|\omega) = 0$ if $i \neq j$.

We will first consider the latter case. For all observations $\omega$, there exists some $k'$ such that $\mathbb{P}(a_{k'}|\omega) > 0$. We then have that for all $i \neq j$, $\sum_\omega \mathbb{P}(\omega|s_i)\mathbb{P}(a_{k'}|\omega)\mathbb{P}(s_j|\omega) = 0$. Because each term in the sum is nonnegative, this is equivalent to the statement that for all $i \neq j$ and all $\omega$, $\mathbb{P}(\omega|s_i)\mathbb{P}(a_{k'}|\omega)\mathbb{P}(s_j|\omega) = 0$. Because $\mathbb{P}(a_{k'}|\omega)$ is positive, this implies that for all $i \neq j$ and for all $\omega$, $\mathbb{P}(\omega|s_i)\mathbb{P}(s_j|\omega) = 0$. This means that if state $s_i$ produces an observation $o$, then $\omega$ cannot be produced by any other reachable state $s_j \neq s_i$, where two states are reachable if there exists a sequence of actions sampled from the policy that enable the agent to reach state $s_i$ from $s_j$ with nonzero probability. In other words, we are in a Block MDP for each component of the state space.

The former case doesn't add anything new. We have that for all $i, k$, $\sum_\omega \mathbb{P}(\omega|s_i)\mathbb{P}(a_k|\omega)(1 - \mathbb{P}(s_i|\omega)) = 0$. Because each term is nonnegative, this is equivalent to $\mathbb{P}(\omega|s_i)\mathbb{P}(a_k|\omega)(\mathbb{P}(s_i|\omega) - 1) = 0$. Because we again have that for all observations there exists an action $a_{k'}$ with nonzero probability, this means we can choose $k = k'$ to find $\mathbb{P}(\omega|s_i) = 0$ or $\mathbb{P}(s_i|\omega) = 1$ for all $\omega, s_i$. This means that either the state $s_i$ does not produce an observation $\omega$, or the observation $\omega$ uniquely determines which state the agent is in.

Lastly, by going backwards through the proof, we see that the converse is also true. If the system is a Block MDP, then $\Pi^S = \Phi W^\Pi$ and so $A_\pi^\lambda = A_\pi^{\lambda'}$.

## D   PROOF OF LEMMA 2 (ZERO $\lambda$-DISCREPANCY)

In this section, we prove Lemma 2, deriving a condition for the $\lambda$-discrepancy to vanish.

Let $A^\lambda = \left(I - \gamma T \left(\lambda \Pi^S + (1 - \lambda)\Phi W^\Pi\right)\right)^{-1}$. $Q^\lambda = Q^{\lambda'}$ iff $W(A^\lambda - A^{\lambda'})R^{SA} = 0$. Expanding $A^\lambda$ and $A^{\lambda'}$ into power series, we have

$$
\begin{aligned}
0 = &\left(WR^{SA} + \gamma WT \left(\lambda \Pi^S + (1 - \lambda)\Phi W^\Pi\right) R^{SA} + \right.\\
&\quad \left. \gamma^2 WT \left(\lambda \Pi^S + (1 - \lambda)\Phi W^\Pi\right) T \left(\lambda \Pi^S + (1 - \lambda)\Phi W^\Pi\right) R^{SA} + \ldots\right)\\
&- \left(WR^{SA} + \gamma WT \left(\lambda' \Pi^S + (1 - \lambda')\Phi W^\Pi\right) R^{SA} + \right.\\
&\quad \left. \gamma^2 WT \left(\lambda' \Pi^S + (1 - \lambda')\Phi W^\Pi\right) T \left(\lambda' \Pi^S + (1 - \lambda')\Phi W^\Pi\right) R^{SA} + \ldots\right)
\end{aligned}
$$

We observe that it is potentially possible to get cases of zero $\lambda$-discrepancy from pairs of terms in this equation cancelling out. Concretely, this means

$$
WT \left(\lambda \Pi^S + (1 - \lambda)\Phi W^\Pi\right) R^{SA} WT\Pi^S R^{SA} - WT\Phi W^\Pi R^{SA} = 0
$$

$$
WT\Pi^S T\Pi^S R^{SA} - WT\Phi W^\Pi T\Phi W^\Pi R^{SA} = 0
$$

$$
\ldots
$$

This can occur in one particularly nice way if $\lambda = 0$ and $\lambda' = 1$. In this case, there is no $\lambda$-discrepancy precisely when

$$
\begin{aligned}
0 = &\left(WR^{SA} + \gamma WT\Pi^S R^{SA} + \gamma^2 WT\Pi^S T\Pi^S R^{SA} + \ldots\right)\\
&- \left(WR^{SA} + \gamma WT\Phi W^\Pi R^{SA} + \gamma^2 WT\Phi W^\Pi T\Phi W^\Pi R^{SA} + \ldots\right)\\
= &\gamma \left(WT\Pi^S R^{SA} - WT\Phi W^\Pi R^{SA}\right)\\
&+ \gamma^2 \left(WT\Pi^S T\Pi^S R^{SA} - WT\Phi W^\Pi T\Phi W^\Pi R^{SA}\right)\\
&+ \ldots
\end{aligned}
\tag{8}
$$

This occurs when each pair of $n$-step return terms cancel, or

$$
WT\Pi^S R^{SA} - WT\Phi W^\Pi R^{SA} = 0
$$

$$
WT\Pi^S T\Pi^S R^{SA} - WT\Phi W^\Pi T\Phi W^\Pi R^{SA} = 0
$$

$$
\ldots
$$

Interpreting this in terms of probabilities, this says that for all initial observations $\omega^0$, initial actions $a^0$, and horizons $T$,

$$
\mathbb{E}(r^T|\omega^0, a^0) = \sum_{\omega^1, \ldots, \omega^T} \mathbb{E}(r^T|\omega^T)\mathbb{P}(\omega^1|\omega^0, a^0) \prod_{t=1}^{T-1} \mathbb{P}(\omega^{t+1}|\omega^t)
\tag{9}
$$

In particular, this applies when $R^{SA}$ is a constant matrix, as in this case, the expected rewards are all equal. Therefore, when $R^{SA}$ is constant, $Q^1 = Q^0$.

Next, we prove that the converse holds for almost all $\gamma$. Consider what happens if $\gamma$ is not fixed with the POMDP, but is instead allowed to vary. In that case, we have that the function $f$ from $\gamma$ to the $\lambda$-discrepancy is an analytic function of $\gamma$:

$$
\begin{aligned}
f : \gamma \mapsto &\gamma \left(WT\Pi^S R^{SA} - WT\Phi W^\Pi R^{SA}\right)\\
&+ \gamma^2 \left(WT\Pi^S T\Pi^S R^{SA} - WT\Phi W^\Pi T\Phi W^\Pi R^{SA}\right)\\
&+ \ldots
\end{aligned}
\tag{10}
$$

Therefore, following an approach similar to Appendix B, we know that either $f$ is everywhere 0, or it is nonzero almost everywhere. This means that either

1. The $\lambda$-discrepancy is 0 for all values of $\gamma$.
2. The $\lambda$-discrepancy is nonzero for almost all values of $\gamma$.

If $f$ is identically 0, then each coefficient of $\gamma$ in the power series expansion of the $\lambda$-discrepancy must be 0, which implies Condition 9. Therefore, the condition is equivalent to the $\lambda$-discrepancy being 0 in this case.

If $f$ is nonzero almost everywhere, then the $\lambda$-discrepancy can only be zero on a set of $\gamma$ of measure 0. Therefore, for almost all $\gamma$, the condition holds if and only if the $\lambda$-discrepancy is actually 0.

## E  MEMORY OPTIMIZATION DETAILS

### E.1  MEMORY-AUGMENTED POMDP

As referenced in Section 4.2, here we will explain how to define a memory-augmented POMDP from a base POMDP $(S, A, T, R, \Omega, \Phi, \gamma)$. Given a set of memory states $M$, we will augment the POMDP as follows:

$$S_M = S \times M$$
$$A_M = A \times M$$
$$\Omega_M = \Omega \times M$$
$$T_M : S_M \times A_M \times S_M \to [0,1], (s_0, m_0) \times (a_0, m_1) \times (s_1, m_2) \mapsto T(s_0, a_0, s_1)\delta_{m_1 m_2}$$
$$R_M : S_M \times A_M \to [0,1], (s_0, m_0) \times (a_0, m_1) \mapsto R(s_0, a_0)$$
$$\Phi_M : S_M \times \Omega_M \to [0,1], (s_0, m_1) \times (\omega_0, m_2) \mapsto \Phi(s_0, \omega_0)\delta_{m_1 m_2}$$
$$\gamma_M = \gamma$$

This augmentation scheme uses the memory states $M$ in three ways: as augmentations of states, actions, and observations. The state augmentation concatenates the environment state $S$ with the agent's internal memory state $M$. Meanwhile, the action augmentation $A_M$ provides the agent with a means of managing its internal memory state. Together, these augmentations allow writing the augmented transition dynamics $T_M$, which are defined so as to preserve the underlying state-transition dynamics $T$ while allowing the agent full control to select its desired next memory state. The observation augmentation $\Omega_M$ provides the agent with additional context with which to make policy decisions, and the observation function $\Phi_M$ preserves the original behavior of the observation function $\Phi$ while giving the agent complete information about the internal memory state.

We define an augmented policy $\pi$ as follows:

$$\pi : \Omega_M \times A_M \to [0,1],$$

which we decompose into two parts, an action policy and a memory function:

$$\pi(a, m'|\omega, m) = \pi_A(a|\omega, m)\mu(m'|a, \omega, m),$$
$$\pi_A : \Omega \times M \times A \to [0,1],$$
$$\mu : \Omega \times M \times A \times M \to [0,1].$$

Note that the memory policy $\mu$ has the same function signature as—and is equivalent to—the state-machine formulation of memory functions introduced in Section 4. This definition of memory functions is convenient because it allows us to express a memory-augmented POMDP as a new POMDP using simple Cartesian products.

Now we can define the value function over the Cartesian product of observations and memories:

$$Q_{\pi_M}^\lambda = W_M \Big( I - \gamma T_M \big( \lambda \Pi^S + (1 - \lambda)\Phi W_M^\Pi \big) \Big)^{-1} R_M, \tag{11}$$

where all quantities (including $W_M$ and $W_M^\Pi$) have been modified to handle the augmented states and observations. We provide pseudocode for taking this memory-Cartesian product of a POMDP in Appendix E.4, Algorithm 3.

### E.2  LAMBDA DISCREPANCY NORM WEIGHTING

The $\lambda$-discrepancy as introduced in Definition 1, and expanded on in Equation 7, contains a weighted norm over the observations and actions of the decision process. There are many choices of norm and weighting scheme. We use an $L^2$ norm to highlight the connection to mean-squared action-value error. For the weighting scheme, we weight actions according to the policy for each observation, and we weight observations uniformly. More precisely, the weighting assigns the $(\omega, a)$ entry the weight $(1, \pi(\omega|a))$. We also considered weighting observations according to their discounted visitation frequency, but found that this led to worse performance during memory optimization.

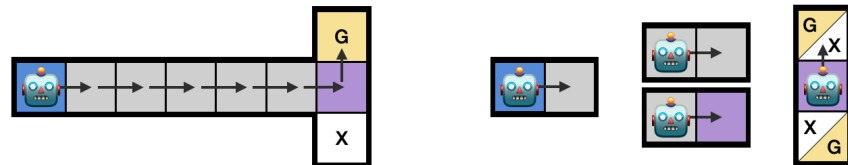

Figure 3: Visualizations of value functions computed using MC (left) and TD (right). MC averages over entire trajectories, so it can associate the blue observation with the upward goal. By contrast, TD computes value by bootstrapping; its value estimates for subsequent observations ignore any prior history.

## E.3 Environments and Experimental Details

### E.3.1 T-Maze Details

We use T-maze with corridor length 5 as an instructive example. The environment has 15 underlying MDP states: one initial state for each reward configuration (reward is either up or down), five for each corridor, one for each junction, and finally the terminal state. There are 5 observations in this environment - one for each of the initial states, a corridor observation shared by all corridor states, a junction observation shared by both junction states, and a terminal observation. The action space is defined by movement in the cardinal directions. If the agent tries to move into a wall, it remains in the current state. From the junction state, the agent receives a reward of $+4$ for going north, and $-0.1$ for going south in the first reward configuration. The rewards are flipped for the second configuration. The environment has a discount rate of $\gamma = 0.9$.

This environment makes it easy to see the differences between MC and TD approaches to value function estimation. We visualize these differences in Figure 3. MC computes the average value for each observation by averaging the return over all trajectories starting from that observation. By contrast, TD averages over the 1-step observation transition dynamics and rewards, and bootstraps off the value of the next observation. For a policy that always goes directly down the corridor and north at the junction, this leads to an average (undiscounted) return for the blue observation of $+4$ with MC and $(4 - 0.1)/2 = 1.95$ with TD.

### E.3.2 Other POMDP Details

All other POMDPs used in the experiments Section 4.2 were taken from pre-defined POMDP definitions (Cassandra, 2003). The only exception is the Tiger environment, where we preserve the underlying environment behavior, but adapt the domain specification to match our formalism such that observations are only a function of state.

The original Tiger domain used a hand-coded initial belief distribution that was uniform over the two states L/R, and did not emit an observation until after the first action was selected. Thereafter, the

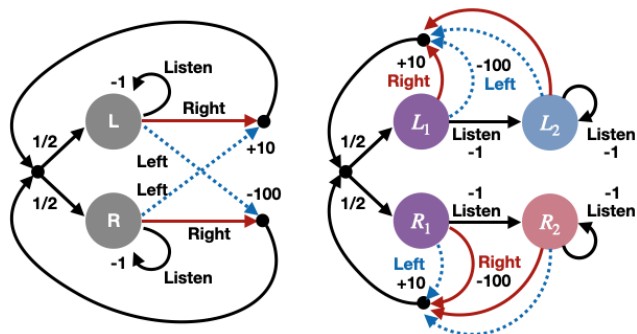

Figure 4: Visualizations of the Tiger POMDP. In the original version (left) the observation function was action-dependent, whereas in our modified version (right) observations only depend on state. The state color for the domain on the right represents the distinct state-dependent observation functions: purple states use the `initial` observation, while the other states are biased towards either `left` (blue) or `right` (red) observations with probability 0.85.

observation function was action-dependent, with state-action pair (`L`, `listen`) emitting observations `left` and `right` with probability 0.85 and 0.15 respectively, and other actions (`L`, `*`) emitting uniform observations and returning to the initial belief distribution. Since our agent does not have access to the set of states, it cannot use an initial belief distribution. To achieve the same behavior, we modified the domain by splitting each state `L`/`R` into an initial state $L_1$/$R_1$ that always emits an `initial` observation, and a post-listening state $L_2$/$R_2$ that uses the 0.85/0.15 probabilities. We visualize these changes in Figure 4. This type of modification is always possible for finite POMDPs and does not change the underlying dynamics.

### E.3.3 ANALYTIC MEMORY LEARNING AND VALUE IMPROVEMENT ALGORITHM

In Algorithm 1, the `policy_improvement` function can be any function which improves a parameterized policy. We consider both the policy gradient (Sutton et al., 2000) algorithm and policy iteration (Howard, 1960). The `randomly_init_n_policies` function returns $n$ randomly initialized policies, and the `select_argmax_ld` function picks the one with the largest $\lambda$-discrepancy. The `memory_improvement` function is defined in Appendix E.3.5 for gradient-based optimization and Appendix E.6 for hill climbing.

We have noticed that larger $\lambda$-discrepancy tends to lead to better memory functions. Although sampling random policies for memory improvement is highly likely to reveal a $\lambda$-discrepancy, it may not be a particularly large $\lambda$-discrepancy. For this reason, we consider many random policies ($n = 400$), and we also consider a memoryless optimal policy (learnt using the chosen policy improvement algorithm), and then use the policy which had maximum $\lambda$-discrepancy as the basis for memory optimization.

### E.3.4 POLICY ITERATION EXPERIMENTS

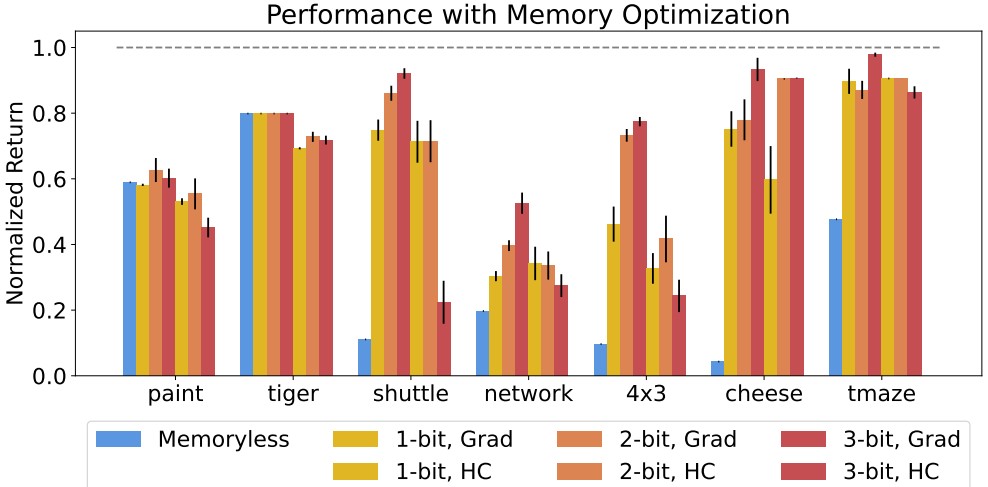

Figure 5: Memory optimization increases normalized return of subsequent policy iteration learning as well. Solid bars denote gradient-based memory optimization (Grad) and hatched bars denote hill climbing (HC). Performance is also normalized between a random policy and belief-state optimal. Error bars are standard error of the mean over 30 seeds.

In this ablation study, we conduct experiments with policy iteration (Howard, 1960) as the policy improvement algorithm. In this setting we also see improved performance over the memoryless policy, and in some cases by a larger margin than the results in Figure 2. Using policy gradient as the policy improvement algorithm improves performance for both memoryless and memory-augmented agents, as compared to using policy iteration. The differences between these results and those of Figure 2 reflect the fact that policy gradient is better able to optimize over stochastic policies, and such policies tend to perform better under partial observability (Sutton and Barto, 2018).

### E.3.5 MEMORY IMPROVEMENT ALGORITHM

In this section we provide pseudocode for the memory-learning algorithm described in Section 4.2 and used in Algorithm 2. This function takes as input a POMDP $\mathcal{P}$ and memory parameters $\theta_\mu$, and minimizes the $\lambda$-discrepancy as defined in Equation 7. This minimization is achieved through a gradient descent update computed using the auto-differentiation package JAX (Bradbury et al., 2018).

---

**Algorithm 2** Memory Improvement (Gradient-Based)

---

**Input:** Fixed policy parameters $\theta_\pi$, where $\Pi = \mathrm{softmax}(\theta_\pi)$, memory parameters $\theta_\mu$, POMDP parameters $\mathcal{P} := (T, R^{SA}, \phi, p_0, \gamma)$, number of improvement steps $n_{\mathrm{steps},\mu}$, learning rate $\alpha \in [0, 1]$

**for** $i = 0$ **to** $n_{\mathrm{steps},M} - 1$ **do**

  *// Augment MDP with memory parameters $\theta_\mu$*

  $\mathcal{P}_{\theta_\mu} \leftarrow$ `expand_over_memory`$(\mathcal{P}, \theta_\mu)$

  *// Calculate MC (no memory augmentation) and TD (with memory augmentation) value functions.*

  $$Q_\pi^1 = W\left(I - \gamma T\Pi^S\right)^{-1} R^{SA}, \; Q_{\pi_\mu}^0 = W_M\left(I - \gamma T_M \Phi_M W_M^\Pi\right)^{-1} R_M$$

  *// Map $\Omega \times M$-space value function back to $\Omega$-space with $p(m \mid o), \forall o \in \Omega$*

  $\hat{Q}_\pi^0 = \sum_{m \in M} p(m|\cdot) Q_{\pi_\mu}^0$

  *// Calculate the $\lambda$-discrepancy*

  $Q_{\pi_\mu}^\lambda = ||\hat{Q}_{\pi_\mu}^0 - Q_\pi^1||_{\pi_\theta, 2}$

  *// Calculate the gradient of $Q_{\pi_\mu}^\lambda$ w.r.t. $\theta_M$, update memory parameters*

  $\theta_\mu \leftarrow$ `update_params`$(\alpha, \theta_\mu, \nabla_{\theta_\mu} Q_{\pi_\mu}^\lambda)$

**end for**

**return** $\theta_\mu$

---

Here, `update_params()` is any gradient-descent-like update, such as stochastic gradient descent, Adam, etc. As a note, all parameters $\boldsymbol{\theta}$ in these experiments are initialized with a Gaussian distribution, with mean 0 and standard deviation 0.5.

### E.4 MEMORY CARTESIAN PRODUCT

In this section, we define the memory-Cartesian product function, `expand_over_memory()`, used by Algorithm 2. This function computes the Cartesian product of the POMDP $\mathcal{P}$ and the memory state space $M$, as described in Appendix E.1.

---

**Algorithm 3** Memory Cartesian Product (`expand_over_memory`)

---

**Input:** Memory parameters $\theta_M$ (with corresponding memory function $M$), POMDP parameters $\mathcal{P} := (T, R^{SA}, O, p_0, \gamma)$, number of memory states $|M|$

*// Repeat reward function for each state over each memory $m \in M$.*

$R_M^{SA} \leftarrow$ `repeat_over_states`$(R^{SA}, |M|)$

*// Calculate transition function cross product.*

$T_M^O \leftarrow$ `einsum`$('ij, jklm \rightarrow iklm', O, M)$

$T_M \leftarrow$ `einsum`$('iljk, lim \rightarrow lijmk')$

*// Calculate observation function cross product. $I_{|M|}$ is the identity function over $|M|$.*

$O_M \leftarrow$ `kron`$(O, I_{|M|})$

*// Finally, calculate the initial state distribution.*

$p_{0,M=0} \leftarrow p_0$

**return** $(T_M, R_M^{SA}, O_M, p_{0,M}, \gamma)$

---

Note that `einsum` is the Einstein summation, and `kron` is the Kronecker product.

### E.5 GRADIENT-BASED EXPERIMENT DETAILS

#### E.5.1 GRADIENT-BASED MEMORY OPTIMIZATION EXPERIMENT DETAILS

For all experiments in Section 4.2, we run memory optimization on the suite of POMDPs with the following hyperparameters. We optimize memory for $n_{\text{steps},M} = 20K$ steps and run policy iteration for $n_{\text{steps},\pi} = 10K$ steps. For all gradient-based experiments, we use the Adam optimizer (Kingma and Ba, 2015).

For the belief-state baselines, solutions were calculated using a POMDP solver from the `pomdp-solve` package (Cassandra, 2003). The performance of the belief-state optimal policy was calculated by iterating all potential initial observations, calculating their corresponding belief states, and taking the dot product between this belief state and the maximal alpha vector for that belief state. This returns a metric comparable to the initial state distribution weighted value function norm, which we use as a performance metric for our memory-augmented agents.

The belief-state solution for the $4 \times 3$ maze was solved using an epsilon parameter of $\epsilon = 0.01$, due to convergence issues with the environment when utilizing POMDP solvers.

### E.6 HILL CLIMBING IMPLEMENTATION DETAILS

---

**Algorithm 4** Memory Improvement (Simulated Annealing)

---

**Input:** Fixed policy parameters $\theta_\pi$, where $\Pi = \text{softmax}(\theta_\pi)$, starting memory parameters $\theta_\mu^0$, TD($\lambda$) parameters $\lambda$ and $\lambda'$, value oracle $Q(\Pi, \lambda, \theta_\mu)$, number of annealing steps $n_{\text{steps}}$, number of random restarts $n_{\text{restarts}}$, annealing temperature bounds $[t_{\min}, t_{\max}]$

$\theta_\mu \leftarrow \theta_\mu^0$
$\Delta \leftarrow \|Q(\Pi, \lambda, \theta_\mu) - Q(\Pi, \lambda', \theta_\mu)\|$
$\theta_\mu^* \leftarrow \theta_\mu$
$\Delta^* \leftarrow \Delta$
**for** $j = 0$ **to** $n_{\text{restarts}} - 1$ **do**
  $\theta_\mu \leftarrow \theta_\mu^0$
  $t_{\text{start}}, t_{\text{end}} \sim \text{Uniform}([t_{\min}, t_{\max}])$ s.t. $t_{\text{start}} \geq t_{\text{end}}$
  $\rho \sim \text{Uniform}([0.1, 0.9])$
  $\alpha \leftarrow \text{DecayRate}(t_{\text{start}}, t_{\text{end}}, \rho, n_{\text{steps}})$
  $\Delta \leftarrow \|Q(\Pi, \lambda, \theta_\mu) - Q(\Pi, \lambda', \theta_\mu)\|$
  **for** $i = 0$ **to** $n_{\text{steps}} - 1$ **do**
    $\theta_\mu' \sim \text{Uniform}(\text{LocalSearchNeighborhood}(\theta_\mu))$
    $\Delta' \leftarrow \|Q(\Pi, \lambda, \theta_\mu') - Q(\Pi, \lambda', \theta_\mu')\|$
    **if** $\Delta' \leq \Delta$ **then**
      $\theta_\mu \leftarrow \theta_\mu'$
      $\Delta \leftarrow \Delta'$
    **else**
      $c \sim \text{Uniform}(0, 1)$
      $T \leftarrow \text{TemperatureSchedule}(t_{\text{start}}, t_{\text{end}}, \alpha, n_{\text{steps}}, i)$
      **if** $c < \exp\left(\frac{-(\Delta' - \Delta)}{T}\right)$ **then**
        $\theta_\mu \leftarrow \theta_\mu'$
        $\Delta \leftarrow \Delta'$
      **end if**
    **end if**
    **if** $\Delta < \Delta^*$ **then**
      $\theta_\mu^* \leftarrow \theta_\pi$
      $\Delta^* \leftarrow \Delta$
    **end if**
  **end for**
**end for**
**return** $\theta_\mu^*$

---

The hill climbing algorithm performs standard simulated annealing over the state space of discrete memory functions in which the energy function to be minimized is defined as the $\lambda$-discrepancy for a given memory function.

Similar to the gradient-based optimization scheme, the initial policy is selected to be the one with the largest $\lambda$-discrepancy from among a set of candidate policies. See Appendix E.3.3 for details.

We define the local search neighborhood for a given deterministic memory function $\mu$ as the set of memory functions that differ from $\mu$ by exactly one edit to the next-memory-state transition table. An edit can modify: either a single observation $\omega_i$ or all observations; either a single action $a_j$ or all actions; either a single memory state $m_k$ or all memory states. We sample uniformly at random from among the set of edits, which, for $|\Omega|$ observations, $|A|$ actions, and $|M|$ memory states, amounts to $(|\Omega|+1) \cdot (|A|+1) \cdot (|M|+1)$ possible edits. The initial memory function $\theta_\mu^0$ is set to the identity function, which always retains the current memory state.

We perform $n_{\text{steps}}$ steps of annealing, then restart the optimization process with a different random seed, for a total of $n_{\text{repeats}}$ trials. Each trial samples temperature parameters $t_{\text{start}}$ and $t_{\text{end}}$ from within the range $[t_{\text{min}}, t_{\text{max}}]$ with $t_{\text{start}} \geq t_{\text{end}}$, as well as a decay rate for exponentially decaying the temperature. The decay rate $\alpha$ is calculated as:

$$\alpha \doteq \frac{\log(t_{\text{max}}/t_{\text{min}})}{n_{\text{steps}} \cdot (1 - \texttt{progress\_fraction\_at\_tmin})},$$

where `progress_fraction_at_tmin` is sampled uniformly from the range $[0.1, 0.9]$. The optimization process outputs the memory parameters that led to the lowest $\lambda$-discrepancy across all annealing steps and random repeats.

# F  CORRELATION WITH VALUE ERROR

Minimizing $\lambda$-discrepancy reduces the value error due to partial observability. To provide additional evidence for this claim, we show in Figure 6 that $\lambda$-discrepancy is positively correlated with value error in every domain we considered.

Here, value error is computed in the following way:

1. Compute $V(s)$ and $V(\omega)$ for the memory-augmented POMDP.
2. Expand $V(\omega)$ to be over $s$, by running the observation function in reverse: $V(\omega) \mathbin{@} \phi^T$, where @ is a matrix product that contracts the $\omega$ dimension. This averages the values of multiple observations if they can be emitted by the same state.
3. Compute the squared difference between the result and $V(s)$, which is a function over $s$.
4. Compute a weighting over s. This can be either uniform or occupancy-weighted. In Fig. 6 we use uniform. The $Q$ version is a weighting over $(s, a)$ and is policy-dependent.
5. Value error is the weighted sum of squared differences.

Note that when $\omega = s$ everywhere, value error will be zero.

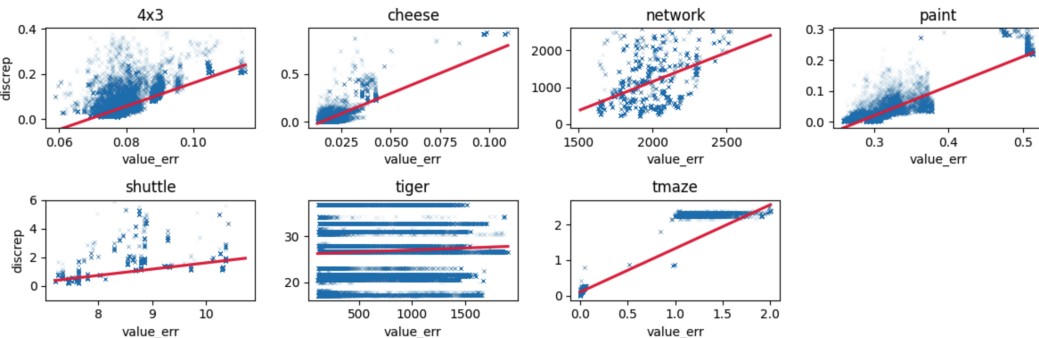

Figure 6: Positive correlation of $\lambda$-discrepancy with value error.

