# OpenReview forum: "Resolving Partial Observability in Decision Processes via the Lambda Discrepancy"
_ICLR.cc/2024/Conference — Submitted to ICLR 2024_

### Official Review · Reviewer_zLef · 2023-10-26

**Soundness:** 1 poor
**Presentation:** 1 poor
**Contribution:** 2 fair
**Rating:** 3
**Confidence:** 4

**Summary:**

For partial observable reinforcement learning problems, the paper introduces $\lambda$-discrepancy as the difference between the TD($\lambda$) returns of two different $\lambda$ values. Several properties of $\lambda$-discrepancy are analyzed and they suggest the discrepancy might serve as a measure of partial observability for a POMDP. Based on the idea, the paper further proposes memory optimization methods by minimizing $\lambda$-discrepancy. Empirical results show performance improvement with memory functions learned by the proposed methods compared with the memoryless case.

**Strengths:**

- The idea of using the difference between TD($\lambda$) returns as a partial observability measure is very interesting, and it has the potential to help construct practical efficient algorithms for partial observable problems.

- The paper attempts to analyze when the $\lambda$-discrepancy would be zero, and the analysis suggests that $\lambda$-discrepancy is zero when the observation has some Markovian properties.

- The paper proposes an approach to learn a memory function for POMDP by minimizing $\lambda$-discrepancy, and it also provides two algorithms based on the approach with different levels of oracles available. Experiments show that learned memory function can indeed provide performance improvement over the memoryless case.

**Weaknesses:**

- The notations are very confusing, and many parts of the technical presentations are very difficult to follow. The products of tensors are presented as regular matrix multiplications without specifying which dimensions are used in the multiplications. Many definitions are not precise, like effective policy over latent state, and some augment representations of some tensors. In the proofs, it is very difficult to follow when the indices are meaningless $i, j, k, l, ...$ without specifying their domains.

- There are issues in many definitions in Section 3.1. As mentioned in the background section, the value functions of partial observable problems generally depend on the agent's entire history. But in Section 3.1, the Q-function is given as a function of the observation and action without any justification. This observation-action Q-function may be some useful approximation, but it is used throughout the paper as the correct value estimation without any discussion. Several variables of conditional probabilities like Pr$(s|\omega)$ are presented as given constants, but they are not provided by the observation model like Pr$(\omega|s) = O(\omega|s)$; the probability Pr$(s_t|\omega_t)$ depends on the policy and the time instant $t$ in general.

- The theoretical analysis in Appendix A seems incorrect. Below the first big equation in Appendix A, it is claimed that one can bootstrap by replacing the last line with a term with $Q(\omega_2, a_2)$ when $n=2$, but the claim means that
$$\sum_{w_2, a_2} P(\omega_2 |s_2) P(a_2|\omega_2) Q(\omega_2, a_2) = \sum_{w_2, a_2} P(\omega_2 |s_2) P(a_2|\omega_2)
\sum_{s_3, r_2} P(s_3|s_2, a_2)P(r_2|s_2, a_2, s_3) r_2$$
This equation does not look correct. Even if what the authors mean is to include all the later terms in $...$, the bootstrap does not seem to hold given the time-homogenous definition of the Q-function unless with further justification.

- There some other likely errors in Appendix A. There is a missing summation over $l$ in the second big equation in Appendix A. In the equation for $Q^\lambda$, $Q_n$ is replaced by the equation derived above, but the inner $Q_n$ in that equation is directly replaced by $Q^\lambda$ without any argument.

- Definition 1 defines $\lambda$-discrepancy by an unspecified norm. It is revealed in Appendix E.2 that the norm is l2, but this is not consistent with the proof in Appendix B where the $\lambda$-discrepancy is a weighted difference between the two Q-functions without squares.

- Even if the some of the analysis is true, they only provide properties when $\lambda$-discrepancy is zero. They still cannot explain whether decreasing $\lambda$-discrepancy can reduce the issue of partial observability in some sense. This kind of property may be explored via numerical experiments, but its not done in the paper.

- In the numerical experiments, the proposed method achieves the optimal performance only in the cheese environment. Since increasing the memory size might lead to optimal policy in principle, the inability to achieve optimal performance with more memory in many environments makes the ability of the proposed methods questionable.

**Questions:**

- Can the authors address the technical issues mentioned above, particularly in Section 3.1 and Appendix A?

---

> ### Author Response · Authors · 2023-11-23
>
> Thank you for your feedback.
>
> $ $
>
> -----
> ### Mathematical presentation
>
> $ $
>
> > - notations are very confusing
> > - The products of tensors are presented as regular matrix multiplications without specifying which dimensions are used in the multiplications
> > - many definitions are not precise, like effective policy over latent state
> > - indices are meaningless i,j,k,l,...
>
> $ $
>
> We have fixed the mathematical presentation in the paper to make it easier to follow. Specifically,
>
> 1. We changed the indices from $i,j,k,l,...$ to indices that indicate their domain, such as $a_1, o_1,$ and $s_1$.
> 2. We have clarified the contractions used in the proof of Appendix A. We kept the matrix multiplications written as-is because they make sense as long as one follows the following guideline, which we included in the original paper: in the main paper equations, each tensor contraction is a single contraction except in the products involving $R^{SA}$ and $T$ on the right hand side, which are double contractions. We also tried writing the contractions with dots, but decided this tensor notation offered the best compromise between clarity and succinctness.
> 3. The effective policy over latent state $\Pi^S$ is an $S \times S \times A$ representation of the matrix $\phi \pi$, where $\phi$ is the $S \times \Omega$ observation function, and $\pi$ is the $\Omega \times A$ observation policy. Formally, $\Pi^S_{s,s’,a} = \delta_{s,s’} \phi_{s,o} \pi_{o,a}$.
>
> $ $
>
> -----
> ### Value definitions
>
> $ $
>
> > The value functions of partial observable problems generally depend on the agent's entire history. But in Section 3.1, the Q-function is given as a function of the observation and action without any justification. This observation-action Q-function may be some useful approximation, but it is used throughout the paper as the correct value estimation without any discussion.
>
> $ $
>
> The MC Q-function is indeed well defined, but it is distinct from the Q-function conditioned on the agent’s full history. We define $Q(\omega,a)$ in the usual way: the expected discounted return starting from observation $\omega$, taking action a, and following the current policy thereafter. This expectation averages over all possible agent histories consistent with reaching observation $\omega$. We do not claim that this is equivalent to either the state value function $Q(s,a)$ or the history value function $Q(h,a)$, but it is nevertheless “correct” in that MC is an unbiased estimator of $Q(\omega,a)$, whereas TD($\lambda$) is not, for any $\lambda < 1$.
>
> $ $
>
> -----
> ### Clarifying conditionals
>
> $ $
>
> > Several variables of conditional probabilities like $Pr(s|\omega)$ are presented as given constants, but they are not provided by the observation model like $Pr(\omega|s)=O(\omega|s)$; the probability $Pr(s_t|\omega_t)$ depends on the policy and the time instant $t$ in general.
>
> $ $
>
> We now describe in Appendix A that $P(s|\omega)$ reflects the average of $P(s_t|\omega_t)$ over all timesteps, weighted by visitation probability and discounted by $\gamma$. This is a well-defined stationary quantity, and it can be computed as follows. First solve the system $Ax = b$ to find the discounted state occupancy counts $x = c(s)$, where $A = (I - \gamma (T^\pi)^\top)$ accounts for the policy-dependent state-to-state transition dynamics $T^\pi$, and $b = P_0$ is the initial state distribution over $s$. Then $P(s|\omega) \propto c(s) * P(\omega|s)$, so we can just multiply these terms together and renormalize.
>
> We will release the code alongside the paper so that these and other such computations will be clear.
>
> $ $
>
> -----
> ### Clarifying bootstrapping
>
> $ $
>
> > Below the first big equation in Appendix A, it is claimed that one can bootstrap by replacing the last line with a term with Q(ω_2,a_2) when n=2... the bootstrap does not seem to hold given the time-homogenous definition of the Q-function unless with further justification.
>
> $ $
>
> Yes, we did mean for all the latter terms in "$\dots$" to be included. We rewrote the equations to make this clearer. We agree that using the time-homogenous Q-function bootstrap as is given here results in a quantity not equal to $\mathbb{E}[G_n|ω_0,a_0]$; that corresponds to the fact that the Q-value estimates are biased. The Q-values are commonly used this way, and doing so does not invalidate the results.

---

> > ### Author Response · Authors · 2023-11-23
> >
> > -----
> > ### Fixed inaccuracies
> >
> > $ $
> >
> > > There is a missing summation over l in the second big equation in Appendix A. In the equation for Q^λ, Q_n is replaced by the equation derived above, but the inner Q_n in that equation is directly replaced by Q^λ without any argument.
> >
> > $ $
> >
> > Thank you for catching that. Our original end result is correct, though our definition was not. The equation that defines $Q_n$ should take as an argument a matrix for bootstrapping $Q$. Then, the equation that previously defined $Q^\lambda$ should instead define the $Q^\lambda$ update rule as $Q\gets (1-\lambda)\sum_{n=1}^{\infty}Q_n(Q)$. We define the matrix $Q^\lambda$ as the fixed point of the $Q^\lambda$ update rule, and then we proceed to compute it. See Appendix A in the updated paper.
> >
> > $ $
> >
> > -----
> > ### Norm
> >
> > $ $
> >
> > > Definition 1 defines λ-discrepancy by an unspecified norm. It is revealed in Appendix E.2 that the norm is l2, but this is not consistent with the proof in Appendix B where the λ-discrepancy is a weighted difference between the two Q-functions without squares.
> >
> > $ $
> >
> > Thank you for noticing this. We define the norm to be a weighted $L^2$ norm, with the weighting dependent on the policy. We have corrected the notation throughout the paper to be consistent with this. The weighted $L^2$ norm is equivalent to taking the dot product between a policy-dependent weight matrix and the difference of the value functions, then taking the $L^2$ norm of that. Because the weight matrix is also an analytic function of $\pi$ and the $L^2$ norm, as a norm, is positive-definite, the normed $\lambda$-discrepancy is 0 if and only if the value function difference is 0; thus the result in Appendix B still holds.
> >
> > $ $
> >
> > -----
> > ### Correlation with value error
> >
> > $ $
> >
> > > Even if the some of the analysis is true, they only provide properties when λ-discrepancy is zero. They still cannot explain whether decreasing λ-discrepancy can reduce the issue of partial observability in some sense. This kind of property may be explored via numerical experiments, but it’s not done in the paper.
> >
> > $ $
> >
> > Minimizing $\lambda$-discrepancy reduces the value error due to partial observability. To provide additional evidence for this claim, we show in Appendix F that $\lambda$-discrepancy is positively correlated with value error in every domain we considered.
> >
> > $ $
> >
> > -----
> > ### Comparison with optimal
> >
> > $ $
> >
> > > In the numerical experiments, the proposed method achieves the optimal performance only in the cheese environment. Since increasing the memory size might lead to optimal policy in principle, the inability to achieve optimal performance with more memory in many environments makes the ability of the proposed methods questionable.
> >
> > $ $
> >
> > The optimal performance is computed by a belief-state planner, which means the agent has access to a perfect environment model *and* enough memory bits to distinguish between all possible belief states. By contrast, our method only requires observable quantities, and works even for agents with limited-memory. Nevertheless, we do see that increasing the memory size improves performance.

---

### Official Review · Reviewer_Rr9Z · 2023-10-30

**Soundness:** 3 good
**Presentation:** 3 good
**Contribution:** 2 fair
**Rating:** 3
**Confidence:** 4

**Summary:**

The paper discusses solving POMDP RL problems through memory augmentation. The authors introduce a $\lambda$-discrepancy, which captures the degree of non-Markovian systems. Based on this property, the authors utilize it as an optimization target to augment the agent's observations based on the memory functions to reduce such a discrepancy. Empirical results are included in the paper to verify the performance.

**Strengths:**

This paper proposes the $\lambda$-discrepancy to measure the discrepancy between two Q-value functions under the same policy but different λ. They further show that this measure is useful for detecting and mitigating partial observability in a POMDP. They then prove several theoretical properties that make the $\lambda$-discrepancy a reasonable optimization objective for learning effective memory functions, which can be used to reduce such a discrepancy. Simulation results also verify the performance of the proposed algorithms.

**Weaknesses:**

This paper mentions that the POMDP problem is inherently complex. While memory augmentation is shown to improve performance, it doesn't directly address the computational complexity of solving POMDPs. The efficiency of the proposed approach in more challenging POMDP scenarios is not fully explored.

It is still not clear to me why we need to reduce the $\lambda$-discrepancy, and at least an improved result should be shown by using this technique compared to using a single $\lambda$.

**Questions:**

- POMDPs are hard to solve due to the large space of the belief states, and thus the algorithms are usually computationally and time-inefficient. However, the complexity of the algorithms is not discussed, and no baselines are included in the experiments.

- Does the algorithm require the full information of the POMDP? If so, we can get the optimal solution with a POMDP solver.

- The condition in Lemma 2 is a sufficient condition; can a necessary condition be provided? That may help in designing the memory function.

- It seems that you approximate Equation (7) with Equation (10), what is the justification for that?

---

> ### Author Response · Authors · 2023-11-23
>
> Thank you for your feedback.
>
> $ $
>
> -----
> ### Complexity
>
> $ $
>
> > While memory augmentation is shown to improve performance, it doesn't directly address the computational complexity of solving POMDPs.
>
> $ $
>
> The computational and sample complexity of computing the $\lambda$-discrepancy are asymptotically the same as computing either of the two Q functions. The algorithm we present is only intended to provide an initial demonstration of the concept; it is not intended to be optimal. Our paper is focused solely on showing (both theoretically and empirically) that minimizing the $\lambda$-discrepancy leads to useful memory functions. The approach is applicable to any POMDP for which we can obtain an accurate observation-based Q function, but obtaining such a Q function is the subject of a large body of work and is outside the scope of this paper.
>
> $ $
>
> -----
> ### Scalability
>
> $ $
>
> > The efficiency of the proposed approach in more challenging POMDP scenarios is not fully explored.
> >
> > $ $
> >
> > POMDPs are hard to solve due to the large space of the belief states, and thus the algorithms are usually computationally and time-inefficient. However, the complexity of the algorithms is not discussed
>
> $ $
>
> (See "Scalability concerns" above.)
>
> $ $
>
> -----
> ### Single-$\lambda$?
>
> $ $
>
> > It is still not clear to me why we need to reduce the $\lambda$-discrepancy, and at least an improved result should be shown by using this technique compared to using a single $\lambda$.
>
> $ $
>
> We are not aware of any method that can learn a memory function from a fixed TD($\lambda$) value function. TD($\lambda$) implicitly makes a Markov assumption for all $\lambda<1$, but there is no way to determine from a single value function whether the Markov assumption is a reasonable one. This is precisely why we need the second $\lambda$. The discrepancy between value functions reliably reveals partial observability and allows us to find a memory function that corrects for it.
>
> $ $
>
> -----
> ### Baselines
>
> $ $
>
> > no baselines are included in the experiments.
>
> $ $
>
> (See "Comparisons to baselines" above.)
>
> $ $
>
> -----
> ### Partial/Full Information
>
> $ $
>
> > Does the algorithm require the full information of the POMDP? If so, we can get the optimal solution with a POMDP solver.
>
> $ $
>
> No, we do not require the full information of the POMDP. The lambda discrepancy can be constructed only from observable quantities. Our method does not require the use of *any* environment information, not even knowledge of the complete *set* of states, let alone the current one. By contrast, belief-state planning methods require an accurate model of the transition dynamics, rewards, observation function, and the full set of states over which the belief has its support. Moreover, belief-state methods assume the agent has enough memory capacity to fully distinguish all possible belief states from each other, whereas our method is effective for limited-memory agents as well. This is why we normalize our performance results for each domain relative to the range from a uniform random policy (y=0) and the unrealistic POMDP-solver solution (y=1).
>
> We only use the environment model in Section 4.2 to show that descending the gradient of the lambda discrepancy leads to useful memory functions. In Section 4.3 we show that using only accurate point estimates of the value function to minimize $\lambda$-discrepancy is as effective as using the  environment model to find gradients.
>
> $ $
>
> -----
> ### Necessary Condition?
>
> $ $
>
> > The condition in Lemma 2 is a sufficient condition; can a necessary condition be provided? That may help in designing the memory function.
>
> $ $
>
> Yes, we have the following necessary condition: Given a fixed policy and $λ_1$, $λ_2$ values, for almost all discount factor values $\gamma$, the converse holds, namely the condition in Lemma 2 implies that there is zero $\lambda$-discrepancy. We have added this to the updated paper, and included the proof in Appendix D.
>
> $ $
>
> -----
> ### Approximation
>
> $ $
>
> > It seems that you approximate Equation (7) with Equation (10), what is the justification for that?
>
> $ $
>
> Thank you for bringing this up. The approximation was not central to any of the claims in the paper, so we have since removed Equation (10) from the work and replaced all experimental results that used the approximation with results computed exactly, following Equation (7). This only affected the results for the gradient-based optimization in Section 4.2, and the new results match or exceed the previous performance.

---

### Official Review · Reviewer_ni5q · 2023-11-01

**Soundness:** 2 fair
**Presentation:** 3 good
**Contribution:** 2 fair
**Rating:** 3
**Confidence:** 3

**Summary:**

This paper proposes the $\lambda$-discrepancy, a theoretical measure for determining the Markovness of observations or memory-estimated Markov states in POMDPs. The measure is based on TD$(\lambda)$.

As a brief summary, TD(0) is based on one-steps returns (i.e. bootstrapping) $Q(s_t,a_t) = r_t + \gamma Q(s_{t+1} a_{t+1})$). TD(1), also known as Monte Carlo estimation, is written as $Q(s_t, a_t) = \sum_{t=0}^\infty \gamma^t r_t$. TD($\lambda$) is a generalization of TD(0) and TD(1). The $\lambda$ parameter in TD($\lambda$) interpolates between TD(0) when $\lambda = 0$ and TD(1) when $\lambda = 1$.

The authors propose that using error between two Q approximations of differing $\lambda$ to measure how non-Markovian a state is. The intuition is that an agent with poor memory will have differing estimations of TD($\lambda_1$), TD($\lambda_2$). The authors' optimization objective is to minimize the discrepancy.

The authors demonstrate their approach on classical POMDPs, showing they can get close-to-optimal performance on certain tasks.

**Strengths:**

- The paper is generally well written
- The paper is well motivated -- a metric denoting the Markovness of a POMDP is a very useful tool
- The approach is novel

**Weaknesses:**

- The experiment section is lacking, only comparing against a memory-free baseline
- The tested environments and models are very low dimensional, and it's not clear such a method would scale to more interesting problems
- If I am understanding this correctly, the proposed metric is flawed in that errors in the Q function (not the state estimator) will result in a $\lambda$-discrepancy. For example, given a perfect state estimator $M^*(o_1, o_2, ..., o_n) = s_n$ and two imperfect Q functions $Q_\theta, Q_\phi$: $ \lVert Q_{\theta}^{\lambda_1}(s_n) - Q_{\phi}^{\lambda_2}(s_n) \rVert > 0$. In this case, I do not see how $\lambda$-discrepancy is a better metric of Markovness than the traditional TD(0) Q learning error.

**Questions:**

- What makes the $\lambda$-discrepancy specific to partial observability? A policy trained on a fully-observable MDP will also have a $\lambda$-discrepancy if the Q function is even slightly less-than-optimal.

---

> ### Author Response · Authors · 2023-11-23
>
> Thank you for your feedback.
>
> $ $
>
> -----
> ### Baselines
>
> $ $
>
> > The experiment section is lacking, only comparing against a memory-free baseline
>
> $ $
>
> (See "Comparisons to baselines" above.)
>
> $ $
>
> -----
> ### Scalability
>
> $ $
>
> > The tested environments and models are very low dimensional, and it's not clear such a method would scale to more interesting problems.
>
> $ $
>
> (See "Scalability concerns" above.)
>
> $ $
>
> -----
> ### TD Error
>
> $ $
>
> > I do not see how $\lambda$-discrepancy is a better metric of Markovness than the traditional TD(0) Q learning error.
> >
> > $ $
> >
> > What makes the $\lambda$-discrepancy specific to partial observability? A policy trained on a fully-observable MDP will also have a $lambda$-discrepancy if the Q function is even slightly less-than-optimal.
>
> $ $
>
> In POMDPs, value error comes from two places: estimation error (as in MDPs) and partial observability. Any method that estimates a value function will have some estimation error, and good value estimation approaches will hopefully drive this error to zero over time. The results we present here show that even if we achieve zero estimation error (which is the case for our value function oracle), there would still be error due to partial observability. Whereas Bellman error is currently the main method we use to reduce the former, it does nothing to reduce the latter. Using the $\lambda$-discrepancy, we are able to reduce the error due to partial observability.
>
> To provide further evidence of this, we show in Appendix F that $\lambda$-discrepancy is positively correlated with value error in every domain we considered.

---

> > ### Comment · Reviewer_ni5q · 2023-11-23
> >
> > > Whereas Bellman error is currently the main method we use to reduce the former, it does nothing to reduce the latter.
> >
> > Really? Isn't the Bellman error just the combination of the two errors you state? If I train an LSTM and a Q function jointly until the Bellman error is zero for all transitions, wouldn't you say that the LSTM has learned to produce a Markov state from the observations, and that the Q function has learned the Q values for all state/action pairs?
> >
> > If we assume we already have a value oracle ($Q^*$), but do not have mappings from observations to states, we could freeze the weights of $Q^*$ and learn an LSTM to map $o_1, o_2, \dots o_n \to s_n$ using Bellman error, no? Why would using the Lambda discrepancy here be more effective than the Bellman error?

---

### Official Review · Reviewer_QHWA · 2023-11-07

**Soundness:** 2 fair
**Presentation:** 3 good
**Contribution:** 3 good
**Rating:** 6
**Confidence:** 5

**Summary:**

This paper studies resolving partial observability through learning memory in non-Markov Decision Processes. The authors propose to use the discrepancy between TD($\lambda$) with different $\lambda$s to measure if the extracted states are Markovian. This discrepancy, termed $\lambda$-discrepency, is then used as an objective to learn memory states. The authors analyze the cases where the $\lambda$-discrepancy vanishes. Lemma 1 introduces a simplest case. Lemma 2 analyzes a particular correlation between the reward and the state-action distribution. Theorem 1 examines that such a specific correlation is very rare. Together they show that it is possible in practice to test with one policy whether the $\lambda$-discrepancy is nonzero for all policies, hereby prove that $\lambda$-discrepency can be an effective learning objective. Concrete algorithms are designed for both analytical estimation and deep learning. Empirical results show that the proposed method can successfully learn memory states in a series of classic POMDP tasks, and achieve better performance than TD methods.

**Strengths:**

Markovianizing nMDPs is a classic problem in the literature of sequential decision-making. The authors propose a novel method to tackle this problem using well-developed tools such as TD($\lambda$).

The quality of this work is good. The strongly analytical narrative makes it convincing. The logic flow is smooth. Proves are solid. The authors did a great job balancing the theoretical formulation and intuition.

The proposed problem and solutions for nMDPs are both relevant. Relaxing the Markovian assumption in sequential decision-making is an active topic recently, see Abel et al. 2021, Janner et al. 2021, Chen et al. 2021., Qin et al. 2023. The author may consider citing these prior works if they haven't done that yet.

Abel et al. On the expressivity of markov reward. NeurIPS 2021.
Janner et al. Offline reinforcement learning as one big sequence modeling problem. NeurIPS 2021.
Chen et al. Decision transformer: Reinforcement learning via sequence modeling. NeurIPS 2021.
Qin et al. Learning non-Markovian Decision-Making from State-only Sequences. NeurIPS 2023.

**Weaknesses:**

I am a bit concerned with the scalability of the proposed method. The empirical results do not include a gradient-based method for model-free implementation of $\lambda$-discrepency. I wonder if the authors would like to discuss the concrete challenges they met in such experiments.

**Questions:**

I happened to have reviewed an earlier version of this work. I am very glad to see the authors have taken some suggestions from the reviewers into account and reframed the paper. However, I also find the authors haven't resolved one of those questions in this revised version.

In the previous version, the proposed method was shown to be struggling to acquire satisfying performances in two tasks. In this version, the authors have shown their success in one of them, Network. However, it seems the result of the other, Hallway, is omitted. I hope the authors would like to provide an explanation in their response.

---

> ### Author Response · Authors · 2023-11-23
>
> Thank you for your feedback, both here and for our previous submission. We agree that this version of the paper is a substantial improvement over the previous draft, and we are grateful for the role that your comments played in making that happen.
>
> $ $
>
> -----
> ### Scalability
>
> $ $
>
> > I am a bit concerned with the scalability of the proposed method.
>
> $ $
>
> (See "Scalability concerns" above.)
>
> $ $
>
> -----
> ### Gradients
>
> $ $
>
> > The empirical results do not include a gradient-based method for model-free implementation of $\lambda$-discrepency. I wonder if the authors would like to discuss the concrete challenges they met in such experiments.
>
> $ $
>
> There are a few potential ways to scale this up, and gradient-based methods are one example. As you saw in our previous draft, the $\lambda$-discrepancy can be computed using neural networks, which means that with the proper architecture, we can learn memory functions model-free via gradient descent. The experiments in our previous draft showed some initial success, but we decided to remove them from this version to focus purely on the tabular setting. We still believe that the neural network direction has promise, but we wanted the first paper on the $\lambda$-discrepancy to tell a theory-driven story. We felt that the engineering challenges associated with including neural networks warranted their own paper and were worried that they would overly complicate the story we are telling here.
>
> Note that scaling up may also be possible gradient-free, via a policy-gradient-style approach; however, our initial investigation into this direction has revealed that this too would warrant its own separate paper, as the derivation is substantially more complicated here than for the MDP setting due to the lack of Markov property.
>
> $ $
>
> -----
> ### Hallway
>
> $ $
>
> > In the previous version, the proposed method was shown to be struggling to acquire satisfying performances in two tasks. In this version, the authors have shown their success in one of them, Network. However, it seems the result of the other, Hallway, is omitted. I hope the authors would like to provide an explanation in their response.
>
> $ $
>
> The Hallway results included in the previous draft were misleading for a few reasons.
>
> First, because the problem is too large for a closed-form POMDP solver, those results were instead normalized relative to the impossible standard of an optimal fully-observable state-based policy. We have since run the SARSOP solver, which does work and is a much more realistic comparison, but normalizing relative to multiple baseline algorithms further complicated the discussion.
>
> Second, we found that the small improvement that we had previously achieved over the memoryless baseline was primarily due to the stochasticity introduced by the earlier stochastic form of memory functions we had investigated. Switching the policy optimization to policy gradient allowed us to better optimize over stochastic policies, and the memoryless policy achieves much better performance.
>
> Third, once policy gradient revealed the optimal memoryless policy, it became clear to us that the Hallway domain simply does not benefit from the small amounts of memory we consider here. The domain involves highly-stochastic, agent-centric observations that must be integrated over multiple time steps, and then remembered across multiple stochastic state transitions, all while tracking more distinctions than are possible with such a small number of bits.

---

### Author Response · Authors · 2023-11-23
**Responses to common questions**

Thank you all, for your helpful comments. We have spent the rebuttal period improving the paper, and have just uploaded the new version (with edits highlighted in magenta).

We will respond to your common questions here, and address individual questions/concerns by replying to each review below.

$ $

-----

$ $

### Comparisons to baselines (reviewers ni5q, Rr9Z)

$ $

We feel that our existing results already make a strong case for the $\lambda$-discrepancy’s usefulness without a baseline algorithm. One benefit of our method is that it supports learning a memory function without changing the policy, and it is unclear if any baseline exists that can support this use case.

Note that the choice of baseline algorithm is not as obvious in our setting as it may initially seem. For example, how should the memory function be learned? One option might be to train the memory function to reduce value error directly, but this requires knowing the value function over ground truth states, and hence is a non-starter for an agent faced with partial observability. Another option is an end-to-end approach, where the memory function is optimized in order to increase the value of the policy. However, the value function depends on both the policy and the memory function. If we force the baseline to learn all of these components at the same time, this is unfair to the baseline, since our method uses a value function oracle. On the other hand, if we do provide it with the value function oracle, it is unclear how to train such a method end-to-end.

If you are aware of a baseline algorithm that is an appropriate comparison for our setting, please let us know!

$ $

-----

$ $

### Scalability concerns (reviewers QHWA, ni5q, Rr9Z)

$ $

The $\lambda$-discrepancy objective is fully compatible with optimization using neural networks; however, to keep the paper focused, we elected not to include our results investigating this direction, despite their initial success. While we agree that large-scale experiments can be useful, we do not feel they would contribute meaningfully to the theoretical understanding we are trying to advance here. We have tried to keep the presentation of the $\lambda$-discrepancy as simple and intuitive as possible. Large experiments tend to require substantial engineering to produce successful results, and these optimizations would likely complicate the otherwise simple presentation.

---

### Meta-Review · Area_Chair_h773 · 2023-12-12

**Metareview:**

This paper proposed a method to identify partial observability and design an algorithm to solve POMDP RL problems through memory augmentation.

The reviews are split. However, all the reviewer raised concerns with regard to their experiment, which is not properly addressed. Moreover, the current presentation has room for improvement. For example, the notation in Lemma can not clearly show the dependency of P(W^{t+1 } | w^t) on the policy. This lack of clarity complicates the reviewers' task of verifying the soundness of the results. In the current form, I tend to recommend rejection but am willing to follow SAC’s suggestion.

———

After discussing with SAC, we reached the "rejection" decision for the following reasons:

1. Weakness in theory. The "almost-all" statement is not particularly informative (despite its occasional appearance in ML theory), since it only states that the bad case consists of a measure-zero set. The tricky situation is that one may get close to the measure-zero set (but not get into it), which results in a rapid degradation of performance in theory. For example, a similar claim exists for LSTD (which the author quoted during discussion), that for almost all $\gamma$ the LSTD matrix is invertible, and early literature used this to justify ignoring the invertibility issue of LSTD matrix. However, even if none-zero, if the singular value of the matrix is very small, the algorithm can still suffer from degenerate performance.

2. Gap between theory and practice. The proposed discrepancy is only used as a stopping rule. There are potentially many other heuristic rules that can achieve similar performances, and it is unclear if the empirical performance is really connected to the theoretical developments.

**Justification For Why Not Higher Score:**

See the above comments for the reason of rejection.

**Justification For Why Not Lower Score:**

N/A

---

### Decision · Program_Chairs · 2024-01-16

Reject